# SIM-SHAPLEY: A STABLE AND COMPUTATIONALLY EFFICIENT APPROACH TO SHAPLEY VALUE APPROXIMATION

## ABSTRACT

Shapley values provide a principled framework for feature attribution in machine learning, but their exponential computational complexity limits practical applications. We propose **SIM-Shapley** (**S**tochastic **I**terative **M**omentum for **Shapley** values), a novel approximation method that reformulates Shapley value computation as a stochastic optimization problem. Our key insight is that momentum-based updates with adaptive mini-batch sampling can significantly accelerate convergence while maintaining theoretical guarantees. We prove that SIM-Shapley achieves linear $Q$-convergence—a stronger guarantee than existing methods—and derive tight variance bounds under mild assumptions. Three stability mechanisms ensure robust performance: $\ell_2$ regularization, negative-sampling detection, and initialization bias correction. Empirically, SIM-Shapley reduces computation time by up to 85% while maintaining or improving attribution quality. Unlike methods requiring specific game formulations, our framework is game-agnostic and extends to general sample average approximation problems. Code is available at https://anonymous.4open.science/r/SIM-Shapley.

## 1 BACKGROUND

The deployment of machine learning in high-stakes domains (Ning et al., 2024) demands not only accurate predictions but also interpretable explanations (Doshi-Velez & Kim, 2017). Among existing interpretability methods for explainable AI (XAI), Shapley Value (SV) (Shapley, 1952) has become a foundational solution (Lundberg & Lee, 2017; Rodemann et al., 2024). SV-based explanations support contrastive explanations, identify corrupted or influential features, and often align well with human intuition (Covert et al., 2020; Molnar, 2025).

However, computing exact Shapley values requires evaluating $\mathcal{O}(d \cdot 2^d)$ feature coalitions, rendering them computationally prohibitive even for moderate dimensionality. Despite recent progress (Jethani et al., 2022; Zhang et al., 2024; Musco & Witter, 2025) in accelerating SV computation, these works remain fundamentally limited: they lack transparency (introducing additional black-box interpreters), generalizability (requiring specific game/imputer configurations), and completeness in implementations (supporting only local or global explanations).

To fill the gap, we introduce SIM-Shapley, a stochastic optimization framework that refines the SV computation. Rather than constraining the method to specific games or architectures, we reformulate SV estimation by iteratively refining estimates via mini-batch sampling and momentum updates (Ruppert, 1988).

A high-level comparison of SIM-Shapley with contemporary SV methods is presented in Table 1, demonstrating the advantages of our proposed method. As shown, SIM-Shapley uniquely provides both game and imputer agnosticism—capabilities absent in existing methods—while maintaining minimal hyperparameter overhead and full interpreter transparency.

Our main contributions include: (1) A new SV estimator with provable linear $Q$-convergence, providing faster convergence than state-of-the-art methods while retaining consistency; (2) Stability mechanisms: $\ell_2$ regularization, detection of negative-sampling events, and correction of initialization bias and (3) A scalable framework compatible with diverse cooperative game formulations, naturally

integrating techniques such as paired sampling (Covert & Lee, 2021), and extensible to a wider class of sample-average-approximation (SAA) objectives.

Table 1: Comparison of SIM-Shapley with state-of-the-art SV-based XAI methods. ✓ indicates full support, ✗ indicates no support.

| Property | FastSHAP (Jethani et al., 2022) | SimSHAP (Zhang et al., 2024) | LeverageSHAP (Musco & Witter, 2025) | **SIM-Shapley** |
|---|---|---|---|---|
| Local/Global Explanations | ✓/✗ | ✓/✗ | ✓/✗ | ✓/✓ |
| Classification/Regression Task | ✓/✗ | ✓/✗ | ✗/✓ | ✓/✓ |
| Interpreter Transparency | ✗ | ✗ | ✓ | ✓ |
| Hyperparameter Overhead | High | High | Minimal | Minimal |
| Game Agnostic | ✗ | ✗ | ✗ | ✓ |
| Imputer Agnostic | ✗ | ✗ | ✗ | ✓ |
| Public Code Available | ✓ | ✗ | ✓ | ✓ |

## 1.1 PRELIMINARIES

### 1.1.1 SHAPLEY VALUE AS WEIGHTED LEAST SQUARES

Let $\mathbf{X} \in \mathbb{R}^{n \times d}$ denote the $d$ features for $n$ samples, and let $D = \{1, 2, \ldots, d\}$ represent their index set. Given a predictive model $f$ for the response $Y$, the performance of any subset $S \subseteq D$ can be expressed through a cooperative game $v(S)$. A restricted model $f_S = f(x_S, X_{S^c})$ uses only features in $S$, with the complement $S^c = D \setminus S$ drawn from the conditional distribution $p(X_{S^c} \mid X_S = x_S)$, yielding *local explanations*.

Several choices for $v(S)$ exist. *SHAP* (Lundberg & Lee, 2017) defines $v(S)$ as the expectation of the prediction from $f_S$, whereas *Shapley Effects* (Owen, 2014) use output variance. In this work, we adopt the prediction-loss game:

$$v(S) = -l(\mathbb{E}[f_S \mid x_S], y) \tag{1}$$

following *SAGE* (Covert et al., 2020) unless otherwise specified. Besides local explanations, *global explanations* arise by treating an input-label pair $U = (X, Y)$ as exogenous and defining a *stochastic cooperative game* as:

$$\mathbf{V}(S) = \mathbb{E}_U[v(S, U)], \tag{2}$$

which captures the expected contribution of a feature subset across the entire data distribution (Covert et al., 2020). This formulation provides a unified framework that encompasses both local and global SV-based methods. The *Shapley value* (SV) $\phi_i$ of feature $i \in D$ is defined as the average marginal contribution of feature $i$ across all subsets $S$ that exclude $i$:

$$\phi_i(v) = \frac{1}{d} \sum_{S \subseteq D \setminus \{i\}} \binom{d-1}{|S|}^{-1} (v(S \cup \{i\}) - v(S)). \tag{3}$$

Lundberg & Lee (2017) recast Shapley estimation as kernel-weighted linear regression, later refined by Covert & Lee (2021). Approximating $v(S)$ with the additive form $u(S) = \beta_0 + \sum_{i \in S} \beta_i$ leads to the weighted least squares objective:

$$\min_{\beta_0, \ldots, \beta_d} \sum_{S \subseteq D} \mu_{\mathrm{Sh}}(S) \left(u(S) - v(S)\right)^2, \qquad \mu_{\mathrm{Sh}}(S) = \frac{d-1}{\binom{d}{|S|} |S| (d - |S|)} \tag{4}$$

where $\mu_{\mathrm{Sh}}(S)$ is the Shapley kernel that yields the optimal solution equal to SV (Charnes et al., 1988; Lundberg & Lee, 2017). Specifically, both the empty set $\emptyset$ and the superset $D$ lead to $\mu_{\mathrm{Sh}}(S)$ diverging to infinity. To satisfy the minimization objective in equation 4, the squared term must be forced to vanish, which introduces the constraint condition stated in equation 5.

Let $\boldsymbol{\beta} = (\beta_1, \ldots, \beta_d)$ denote the optimal coefficients (i.e., the minimizer) for equation 4; these are the estimated SV. To represent subsets more conveniently, map each subset $S \subseteq D$ to a binary vector $z \in \{0, 1\}^d$, with $z_i = 1 \Leftrightarrow i \in S$. With a slight abuse of notation, write $v(z) = v(S)$ and $\mu_{\mathrm{Sh}}(z) = \mu_{\mathrm{Sh}}(S)$ for the subset $S = \{i : z_i = 1\}$. We then define a sampling distribution $p(z) \propto \mu_{\mathrm{Sh}}(z)$ over binary vectors $z$ satisfying $0 < \mathbf{1}^\top z < d$; otherwise, we set $p(z) = 0$.

Under this formulation, estimating SV reduces to solving the following constrained optimization problem (Covert & Lee, 2021):

$$\min_{\beta_1,\ldots,\beta_d} \sum_z p(z)\left(v(\mathbf{0}) + z^\top \boldsymbol{\beta} - v(z)\right)^2 \quad \text{s.t.} \quad \mathbf{1}^\top \boldsymbol{\beta} = v(\mathbf{1}) - v(\mathbf{0}). \tag{5}$$

Since the squared term vanishes for $z = \mathbf{0}$ and $z = \mathbf{1}$, these configurations can be excluded from sampling.

### 1.1.2  MONTE CARLO ESTIMATION AND PRACTICAL CHALLENGES

Evaluating equation 5 on all $2^d$ subsets is computationally prohibitive. Instead, we draw $m$ i.i.d. samples $z_i \sim p(z)$ and their values $v(z_i)$, yielding the Monte-Carlo surrogate:

$$\min_{\beta_1,\ldots,\beta_d} \frac{1}{m}\sum_{i=1}^m \left(v(\mathbf{0}) + z_i^T \boldsymbol{\beta} - v(z_i)\right)^2 \quad \text{s.t.} \quad \mathbf{1}^T \boldsymbol{\beta} = v(\mathbf{1}) - v(\mathbf{0}). \tag{6}$$

Covert & Lee (2021) derive a closed-form solution via the Karush–Kuhn–Tucker (Boyd & Vandenberghe, 2004) (KKT) conditions and proved consistency as $m \to \infty$. In practice, however, only a finite number of $z_i$ are available, which may lead to a singular system in certain regions of the solution space and cause the optimization to collapse. Moreover, naively averaging batch estimates yields high variance and fluctuation due to insufficient coverage of the subset space, leading to numerical instability and optimization failure.

### 1.1.3  FROM SAMPLE AVERAGE APPROXIMATION TO STOCHASTIC ITERATION

The constrained problem in equation 5 and its sampled version in equation 6 follow the Sample Average Approximation (SAA) paradigm (Shapiro & Wardi, 1996): we replace the expectation in equation 5 by the empirical average in equation 6. Under standard convexity and regularity conditions, the SAA solution converges almost surely as $m \to \infty$ (Fu, 2006; Kim et al., 2015). However, large $m$ can be impractical due to computational constraints, and not all samples are equally informative for improving the estimator.

Therefore we turn to Stochastic Approximation (SA) methods (Hannah, 2015; Shapiro, 1996), which update the solution incrementally using only a mini-batch at each step. This avoids repeatedly solving the full SAA problem and can lead to faster convergence in practice. As pointed out by Royset & Szechtman (2013), SA can outperform SAA in convergence rate, regardless of whether the underlying solver is sublinear, linear, or superlinear.

## 2  PROPOSED METHOD: SIM-SHAPLEY

### 2.1  STOCHASTIC ITERATION WITH MOMENTUM

Following the stochastic optimization formulation discussed in Section 1.1.3, we reformulate the KernelSHAP approximation (Lundberg & Lee, 2017) in equation 6 as an iterative procedure to update estimates $\boldsymbol{\beta}^{(n)}$ using an *exponential moving average* (EMA) scheme:

$$\min_{\beta_1^{(n+1)},\ldots,\beta_d^{(n+1)}} \frac{1}{m}\sum_{i=1}^m \left(v(\mathbf{0}) + z_i^T \boldsymbol{\beta}^{(n+1)} - v(z_i)\right)^2 + \lambda \left\|\boldsymbol{\delta}^{(n+1)}\right\|_2^2 \tag{7a}$$

$$\text{s.t.} \quad \mathbf{1}^T \boldsymbol{\beta}^{(n+1)} = v(\mathbf{1}) - v(\mathbf{0})$$

$$\boldsymbol{\beta}^{(n+1)} = t\boldsymbol{\beta}^n + (1-t)\boldsymbol{\delta}^{(n+1)} \tag{7b}$$

where $0 < t < 1$ is a fixed momentum parameter. $\boldsymbol{\delta}^{(n)}$ represents the sampling information obtained at the $n$-th iteration, which can be interpreted as a form of momentum. The updated estimator $\boldsymbol{\beta}^{(n)}$ is constructed as a weighted combination of the previous estimate $\boldsymbol{\beta}^{(n-1)}$ and $\boldsymbol{\delta}^{(n)}$. In addition, $\boldsymbol{\beta}^{(n)}$ shall satisfy the constraint in equation 5 to preserve the efficiency property of SV, as reflected in the definition of $u(S)$ (equation 4). Although $\boldsymbol{\delta}^{(n)}$ can also be adopted as the optimization variable owing to its linearity with $\boldsymbol{\beta}^{(n)}$, the optimization in equation 7 is performed with respect to $\boldsymbol{\beta}^{(n)}$ to preserve notational consistency in Section 1.1.

To control the variance of $\boldsymbol{\delta}^{(n)}$ under limited sampling (e.g., when $\{z_1, \ldots, z_m\}$ contains many duplicate subsets), we introduce an $\ell_2$ regularization penalty with coefficient $\lambda > 0$. This penalty ensures strict convexity of the objective function at each iteration and guarantees the existence and uniqueness of a closed-form solution. Intuitively, the penalty stabilizes the updates by discouraging large deviations of $\boldsymbol{\delta}^{(n)}$ from $\boldsymbol{\beta}^{(n-1)}$.

For notational simplicity, we do not distinguish between $v(S)$ in equation 1 and $\mathbf{V}(S)$ in equation 2, as replacing $v(S)$ with $\mathbf{V}(S)$ allows $\boldsymbol{\beta}^{(n+1)}$ in SIM-Shapley to be interpreted as an estimator of the global SV explanation. To approximate $\mathbf{V}(S)$, we construct a *Global Reference Set* by sampling from the original dataset, thereby approximating the distribution $p(U)$ and enabling the evaluation of $\mathbb{E}_U[v(S, U)]$ for each cooperative game.

As discussed in Section 1.1.1, to approximate the conditional distribution $p(X_{S^c} \mid X_S = x_S)$, we handle the missing features $X_{S^c}$ by marginalizing them using their joint marginal distribution. Specifically, we build a *Background Set* from the data and use it to impute the removed variables. Since our focus is on estimating the SV rather than the underlying games themselves, this treatment does not affect the convergence or correctness of the iterative procedure.

Building on these ideas, we propose the SIM-Shapley algorithm, together with its local explanation version described in Algorithm 1. The global version is provided in the Appendix C. We apply the KKT conditions to equation 7, leveraging the convexity of the objective function and the linearity of the constraint. Given a sampled batch $(z_1, \ldots, z_m)$, the associated Lagrangian function, with Lagrange multiplier $\nu \in \mathbb{R}$, is defined as follows:

$$\mathcal{L}(\boldsymbol{\delta}, \nu) = \frac{1}{m} \sum_{i=1}^{m} \left( v(\mathbf{0}) + z_i^\top \boldsymbol{\beta}^{(n+1)} - v(z_i) \right)^2 + \lambda \|\boldsymbol{\delta}^{(n+1)}\|_2^2 + \nu \left( \mathbf{1}^\top \boldsymbol{\delta}^{(n+1)} - c \right). \quad (8)$$

where we further introduce the following shorthand notations:

$$A = \frac{1}{m} \sum_{i=1}^{m} z_i z_i^\top, \qquad \bar{A} = A + \lambda I, \qquad \bar{b} = \frac{1}{m} \sum_{i=1}^{m} z_i \left( v(z_i) - v(\mathbf{0}) \right), \qquad c = v(\mathbf{1}) - v(\mathbf{0}).$$

Solving the KKT conditions yields the closed-form update for $\boldsymbol{\delta}^{(n+1)}$:

$$\boldsymbol{\delta}^{(n+1)} = \frac{1}{1-t} \bar{A}^{-1} \left[ \left( \bar{b} - tA\boldsymbol{\beta}^{(n)} \right) + \mathbf{1} \cdot \frac{c - t\mathbf{1}^\top \boldsymbol{\beta}^{(n)} - \mathbf{1}^\top \bar{A}^{-1} \left( \bar{b} - tA\boldsymbol{\beta}^{(n)} \right)}{\mathbf{1}^\top \bar{A}^{-1} \mathbf{1}} \right] \quad (9)$$

## 2.2 VARIANCE AND CONVERGENCE

We further analyze the variance and convergence behavior of the SIM-Shapley estimator $\boldsymbol{\beta}^{(n)}$ with respect to the ideal SV vector $\boldsymbol{\beta}$. Throughout this paper, we treat the KernelSHAP estimator in equation 6 as the ground truth, consistent with prior works demonstrating that KernelSHAP and its variants are either unbiased or nearly unbiased (Covert & Lee, 2021; Covert et al., 2020; Chen et al., 2022; Lundberg & Lee, 2017). Since SIM-Shapley is designed to approximate KernelSHAP through a stochastic iterative scheme with regularization, any potential bias is empirically negligible with a tiny $\lambda$, as confirmed by Figure 1 and our experiments in Section 4.

The total mean squared error of $\boldsymbol{\beta}^{(n)}$ admits the standard decomposition:

$$\mathrm{MSE}\big(\boldsymbol{\beta}^{(n)}\big) := \mathbb{E}\big[\|\boldsymbol{\beta}^{(n)} - \boldsymbol{\beta}\|_2^2\big] = \underbrace{\mathbb{E}\big[\|\boldsymbol{\beta}^{(n)} - \mathbb{E}[\boldsymbol{\beta}^{(n)}]\|_2^2\big]}_{\text{Variance}} + \underbrace{\|\mathbb{E}[\boldsymbol{\beta}^{(n)}] - \boldsymbol{\beta}\|_2^2}_{\text{Bias}^2}. \quad (10)$$

Below we analyze SIM-Shapley's variance and convergence across iterations, as these determine practical performance in typical settings with negligible bias.

**Theorem 1** (Variance Contraction). Let $\hat{\boldsymbol{\beta}}$ denote the estimator obtained from KernelSHAP in equation 6 with $m$ samples of $z$, and let $\boldsymbol{\beta}^{(n)}$ be the SIM-Shapley iterates with fixed momentum $t \in (0, 1)$. As $m \to \infty$, for fixed momentum $t \in (0, 1)$ and any feature index $i$ and any iteration $n \geq 1$,

$$\mathrm{Var}(\beta_i^{(n)}) \leq (1 - t)^2 \, \mathrm{Var}(\hat{\beta}_i).$$

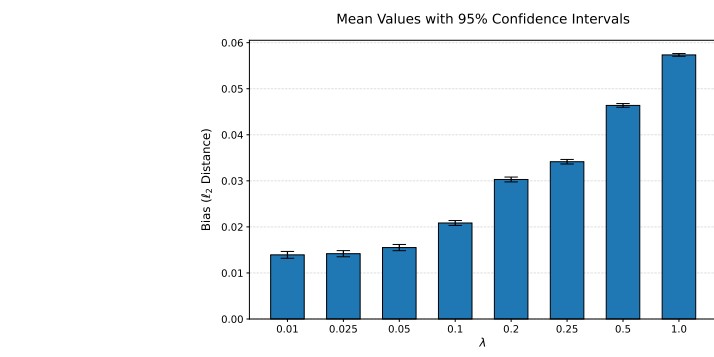

Figure 1: Bias-regularization trade-off for SIM-Shapley on the Bank dataset. Bias represents the $\ell_2$ distance from ground-truth SV as $\lambda$ varies from 0 (no regularization) to 1.

A detailed proof of Theorem 1 is given in Appendix B.1. In essence, each EMA update in SIM-Shapley reduces the variance of feature $i$ by at least a factor of $(1-t)^2$ relative to the vanilla SAA estimator $\hat{\beta}_i$. Because the bias remains comparable to KernelSHAP, this geometric variance contraction provides a principled way to decide when to stop iterating.

Concretely, we monitor the largest standard error across all coordinates and compare it to the span of the current Shapley estimates. Denoting by $\sigma_i^{(n)} = \sqrt{\frac{\mathrm{Var}(\beta_i^{(n)})}{n}}$, the standard error of feature $i$ at iteration $n$, we declare convergence once

$$\max_i \sigma_i^{(n)} \; < \; \epsilon \left( \max_i \beta_i^{(n)} - \min_i \beta_i^{(n)} \right), \tag{11}$$

where $\epsilon$ is a small tolerance (e.g. 0.025). This criterion ensures that the residual uncertainty in any coordinate is a negligible fraction of the overall range of SV. To track $\sigma_i^{(n)}$ online with negligible extra memory, we employ Welford's algorithm (Welford, 1962) within Algorithm 1. Additionally, we provide an empirical analysis on $\epsilon$ corresponding to the ground truth bias in Appendix E.3.

Due to the strict convexity of the underlying quadratic objective and the affine constraint structure, SIM-Shapley admits guaranteed convergence. We now characterize its rate of convergence in terms of the $Q$-linear convergence criterion.

**Theorem 2.** As $m \to \infty$, SIM-Shapley exhibits a linear Q-convergence rate, i.e.,

$$\|\boldsymbol{\beta}^{(n)} - \boldsymbol{\beta}^*\| = \mathcal{O}(\rho(G)^n),$$

where $\boldsymbol{\beta}^*$ is the optimal point of SIM-Shapley, $\rho(G) = t \cdot \frac{\lambda}{\alpha+\lambda}$, and $\alpha > 0$ denotes the minimal eigenvalue of $A$.

Since $0 < t < 1$ and $0 < \frac{\lambda}{\alpha+\lambda} < 1$, $\rho = t \frac{\lambda}{\alpha+\lambda}$ satisfies $\rho < 1$, guaranteeing linear convergence of SIM-Shapley.

Theorem 2 reveals a fundamental trade-off between convergence speed and bias, governed by the choice of $t$ and $\lambda$. The detailed proof is in Appendix B.2. In practice, our objective is accurate SV approximation rather than exact fixed-point attainment, so we continue to use the variance-based stopping rule of equation 11 for early termination.

# 3 Stability Improvement

In this section, we introduce two techniques to enhance the convergence stability of SIM-Shapley. By combining these methods, we propose **Stable-SIM-Shapley** (see Appendix D for complete pseudocode), an improved variant with more robust convergence behavior.

## 3.1 Preventing Negative Sampling

As shown in Section 2.2, the convergence of $\boldsymbol{\beta}^{(n+1)}$ is driven by the variance (covariance) of the update $\boldsymbol{\delta}^{(n+1)}$. In practice, the convergence of SIM-Shapley can stall or even reverse when a mini-

---

**Algorithm 1 SIM-Shapley** (Local Explanation)

---

**Input:** Instance $(x, y)$, background set $(X_{\text{BK}}, Y_{\text{BK}})$, subset sample size $m$, number of iterations $T$
**Output:** Final estimate $\boldsymbol{\beta}^{(n)}$
1: Initialize $\boldsymbol{\beta}^{(0)} = \mathbf{0}$; $Mean_0 = \mathbf{0}$; $S_0 = \mathbf{0}$; precompute $v(\mathbf{1})$ and $v(\mathbf{0})$ for the target instance
2: **for** $n = 1$ to $T$ **do**
3:     Independently sample $\{z_j\}_{j=1}^m \sim p(z)$
4:     For each $z_j$, compute $v(z_j)$ by marginalizing missing features using the background set
5:     Solve equation 7 to obtain $\boldsymbol{\delta}^{(n)}$ and update $\boldsymbol{\beta}^{(n)}$
6:     Update running variance $Var_n$ of $\boldsymbol{\beta}^{(n)}$ using Welford's algorithm:
7:        $\Delta \leftarrow \boldsymbol{\beta}^{(n)} - Mean_{n-1}$;  $Mean_n \leftarrow Mean_{n-1} + \Delta/n$
8:        $S_n \leftarrow S_{n-1} + \Delta \cdot (\boldsymbol{\beta}^{(n)} - Mean_n)$;  $Var_n \leftarrow S_n/(n-1)$
9:     **if** convergence criterion satisfied **then**
10:        **break**
11:     **end if**
12: **end for**
13: **return** $\boldsymbol{\beta}^{(n)}$

---

batch contains many low-quality or duplicate coalitions—what we call *negative sampling*. Such batches produce high-variance updates $\boldsymbol{\delta}^{(n+1)}$, slowing overall convergence.

To detect and avoid these harmful updates, we compare the update variance to the current iterate variance. Specifically, we compute

$$r = \frac{\left\|\text{Var}(\boldsymbol{\delta}^{(n+1)})\right\| - \left\|\text{Var}(\boldsymbol{\beta}^{(n)})\right\|}{\left\|\text{Var}(\boldsymbol{\beta}^{(n)})\right\|} \leq \xi, \tag{12}$$

and if $r > \xi, \quad 0 < \xi < 1$, we reject the batch and resample. In our experiments, $\xi \in \{0.30, 0.35\}$ generally works well (see empirical experiment in Appendix E.3). This incurs minimal cost: one additional mini-batch draw, since variances are already tracked for convergence detection. This check significantly reduces high-variance updates, improving convergence.

## 3.2 CORRECTING INITIALIZATION BIAS

The update rule in our algorithm follows an EMA scheme. However, such schemes are sensitive to initialization and typically suffer from *initial bias*. Specifically, during early iterations, $\boldsymbol{\beta}^{(n)}$ is overly influenced by $\boldsymbol{\beta}^{(0)}$, which holds a disproportionately large weight.

To address this, we initialize $\boldsymbol{\beta}^{(0)} = 0$ and apply a bias correction technique similar to that used in the Adam optimizer (Kingma & Ba, 2017). This leads to a dynamically adjusted update rule, where the weight factor $t$ varies over iterations. The closed-form solution in equation 9 can be adjusted accordingly due to linearity (see Appendix D). The bias-corrected update rule is given by:

$$\boldsymbol{\beta}^{(n+1)} = \frac{t\boldsymbol{\beta}^{(n)}}{1 - t^{n+1}} + \frac{(1-t)\boldsymbol{\delta}^{(n+1)}}{1 - t^{n+1}}. \tag{13}$$

# 4 EXPERIMENTS

We evaluate SIM-Shapley against state-of-the-art methods to assess: (1) accuracy-efficiency tradeoffs, (2) robustness across task types and dimensionalities, and (3) impact of stability mechanisms.

## 4.1 BASELINE METHODS AND EXPERIMENTAL SETUP

We compare against four established methods, each representing different computational approaches: **FastSHAP** (Jethani et al., 2022) uses neural networks to approximate both the conditional distribution and Shapley values, enabling single-forward-pass inference after expensive upfront training. We test both the original architecture and a reduced variant (FastSHAP*) with half the parameters. **Leverage-SHAP** (Musco & Witter, 2025) employs statistical leverage scores to improve sampling efficiency

Table 2: **Local explanation comparison on Census dataset (binary classification)**. All times in milliseconds (ms). LeverageSHAP is incompatible with classification tasks in its current implementation. FastSHAP* denotes the reduced-parameter version of FastSHAP.

| Method | Metric | Sampling Size[2] | | | | | | | | | |
|--------|--------|------|------|------|------|------|------|------|------|------|------|
| | | 64 | 256 | 512 | 768 | 1024 | 1536 | 2048 | 2304 | 2816 | 3840 |
| SIM-Shapley | Bias | 0.016 | 0.016 | 0.016 | 0.010 | 0.009 | 0.008 | 0.008 | 0.008 | 0.008 | 0.008 |
| | Time | 6.23 | 6.07 | 6.38 | 14.03 | 13.13 | 18.51 | 16.99 | 22.53 | 29.51 | 34.03 |
| KernelSHAP | Bias | 0.029 | 0.029 | 0.028 | 0.020 | 0.019 | 0.016 | 0.015 | 0.013 | 0.012 | 0.010 |
| | Time | 7.04 | 5.74 | 8.01 | 13.88 | 13.05 | 15.40 | 22.45 | 32.41 | 34.19 | 40.08 |
| FastSHAP | Bias | 0.031 | | | | | | | | | |
| | Time | 76340 | | | | | | | | | |
| FastSHAP* | Bias | 0.065 | | | | | | | | | |
| | Time | 72210 | | | | | | | | | |

but is limited to mean-value imputation and specific task types. **KernelSHAP** (Lundberg & Lee, 2017) serves as the benchmark for local explanations. As KernelSHAP is asymptotically unbiased, we approximate the ground truth by running KernelSHAP with a sufficiently small convergence threshold, following Zhang et al. (2024); Jethani et al. (2022). **SAGE** (Covert et al., 2020) serves as the benchmark for global explanations. SimSHAP (Zhang et al., 2024) is excluded due to lack of public implementation; however, as it closely parallels FastSHAP's approach, our FastSHAP results provide a reasonable proxy.

**Experimental Protocol.** All experiments use consistent evaluation protocols: (1) Ground truth Shapley values (local explanation) are approximated using KernelSHAP with convergence threshold $\epsilon = 0.025$; (2) Bias is measured as $\ell_2$ distance from ground truth; (3) For fair comparison, all compatible methods use identical imputation strategies; (4) Results are averaged over 100 independent runs[1]. Additional implementation details are in provided in Appendix E.

**Time Measurement.** To ensure fair comparison across methods with different computational structures, we measure time as follows: (i) For FastSHAP, we report the one-time neural network training cost for both imputer and explainer, as inference requires only a negligible forward pass; (ii) For KernelSHAP and SIM-Shapley, we report the estimation time using Algorithm 1 and its corresponding version for KernelSHAP, as both do not require explainer training; (iii) For LeverageSHAP, we report estimation time only, as it uses mean imputation without requiring training. All methods use the same imputer architecture when applicable to isolate the effect of imputer.

### 4.2 LOCAL EXPLANATIONS

#### 4.2.1 CLASSIFICATION TASKS

Table 2 presents results on the Census dataset for binary classification. Several key findings emerge: (1) SIM-Shapley achieves consistently lower bias than KernelSHAP across all sampling sizes while requiring comparable or less computation time; (2) FastSHAP's bias remains fixed at 0.031 regardless of sampling size, as it produces deterministic outputs from its trained neural networks; (3) Reducing FastSHAP's architecture complexity (FastSHAP*) doubles the bias while providing minimal training time reduction, highlighting the method's sensitivity to architectural choices. Similar patterns are observed on the Credit dataset (Appendix E.4).

#### 4.2.2 REGRESSION TASKS

Table 3 presents results on the Airbnb dataset for regression tasks, revealing distinct performance patterns from classification. SIM-Shapley maintains competitive performance, achieving bias within 15% of KernelSHAP while reducing computation time by up to 30% at larger sampling sizes. On

---

[1]Hardware configuration: 12th Gen Intel Core i5-12600KF CPU with NVIDIA RTX 4060 GPU.

[2]Since not all SV methods are compatible with the convergence detection criterion in equation 11, we choose multiple fixed sampling size and compare the resulting bias to ensure fair comparisons.

Table 3: **Local explanation comparison on Airbnb dataset (regression).** All times in milliseconds (ms). FastSHAP is incompatible with regression tasks in its current implementation.

| Method | Metric | Sampling Size | | | | | | | | | |
|--------|--------|------|------|------|------|------|------|------|------|------|------|
| | | 64 | 256 | 512 | 768 | 1024 | 1536 | 2048 | 2304 | 2816 | 3840 |
| SIM-Shapley | Bias | 6.43 | 6.43 | 6.43 | 3.32 | 3.34 | 2.61 | 2.61 | 2.68 | 2.73 | 2.77 |
| | Time | 7.80 | 9.18 | 8.82 | 14.83 | 14.64 | 21.72 | 28.48 | 34.47 | 40.52 | 54.24 |
| KernelSHAP | Bias | 5.94 | 5.55 | 5.53 | 3.96 | 3.99 | 3.54 | 2.90 | 2.48 | 2.28 | 2.08 |
| | Time | 10.62 | 7.22 | 10.84 | 21.27 | 22.45 | 28.69 | 35.35 | 51.47 | 54.63 | 81.50 |
| LeverageSHAP | Bias | 16.19 | 7.39 | 5.68 | 4.32 | 3.72 | 2.93 | 2.65 | 2.45 | 2.26 | 1.98 |
| | Time | 1.88 | 6.52 | 12.99 | 25.68 | 30.92 | 54.96 | 62.50 | 79.57 | 96.88 | 122.85 |

the contrary, LeverageSHAP demonstrates poor efficiency-accuracy tradeoffs: despite achieving the lowest bias at large sampling sizes (1.98 at 3840), it requires $2\times$ the computation time of SIM-Shapley and exhibits extreme bias at smaller sampling size (16.19 at 64).

### 4.2.3 IMAGE DATA APPLICATIONS

On MNIST image data (784 pixel features), SIM-Shapley demonstrates superior scalability (Table 4): 62% faster than KernelSHAP (8.53 seconds vs. 22.47 seconds) with comparable accuracy. Fast-SHAP's explainer training exceeds 11 minutes—more than double imputer training—while yielding $3.5\times$ worse bias, highlighting its poor accuracy-efficiency tradeoff in higher dimensions.

Table 4: **Average runtime and bias of baseline methods on the MNIST dataset.** All times are reported in seconds (s). The estimation time of FastSHAP is negligible, as it involves only a single forward pass of a neural network. For fair comparison, we fix the same imputer for all three methods. LeverageSHAP is incompatible with classification tasks in its current implementation.

| Method | Imputer Training (s) | Explainer Training (s) | Estimation Time (s) | Bias |
|--------|---------------------|------------------------|---------------------|------|
| SIM-Shapley | | *None* | 8.53 | 0.007 |
| KernelSHAP | 314.21 | *None* | 22.47 | 0.006 |
| FastSHAP | | 703 | *Trivial* | 0.025 |

### 4.3 GLOBAL EXPLANATIONS AND STABILITY ANALYSIS

Having demonstrated SIM-Shapley's advantages for local explanations, we now evaluate: (1) its performance on global explanations, and (2) the impact of our stability mechanisms (Stable-SIM-Shapley) across both local and global tasks.

### 4.3.1 GLOBAL EXPLANATION PERFORMANCE

Table 5 compares SIM-Shapley variants against SAGE, the standard for global explanations. Across all datasets, both SIM-Shapley variants achieve 60-75% reduction in computation time while maintaining >95% correlation with SAGE estimates. The stable variant Stable-SIM-Shapley provides additional efficiency gains (up to 31% faster on Credit dataset) through improved convergence properties, without sacrificing accuracy.

### 4.3.2 IMPACT OF STABILITY MECHANISMS

To further evaluate the contribution of our stability mechanisms, Table 6 presents local explanation performance of SIM-Shapley variants. Both achieve excellent agreement with KernelSHAP (correlations $> 0.91$) while providing substantial speedups ($1.5 - 6\times$faster). The Stable variant shows

---

[3]We assess consistency using Pearson's correlation coefficient, as the Shapley estimates from different methods exhibit approximately linear relationships (see Appendix E.1 for details and visualizations).

Table 5: **Global explanation performance.** Running times (seconds) for computing dataset-level SV using convergence criterion from equation 11 ($\epsilon = 0.025$). Consistency measured as Pearson correlation with SAGE baseline. Bold indicates best running time among SIM-Shapley variants.

| Data | Model | Avg. Running Time (s) | | | Avg. Consistency[3] | |
|---|---|---|---|---|---|---|
| | | SIM-Shapley | Stable-SIM-Shapley | SAGE | SIM-Shapley | Stable-SIM-Shapley |
| Bike | XGBoost | 7.95 | **7.56** | 21.66 | 0.9958 | 0.9966 |
| Credit | CatBoost | 23.88 | **17.59** | 61.39 | 0.9654 | 0.9675 |
| Bank | CatBoost | 46.39 | **37.16** | 171.39 | 0.9529 | 0.9518 |
| Airbnb | MLP | 7.05 | **4.87** | 32.80 | 0.9588 | 0.9659 |

mixed results: it improves efficiency for some tasks (Credit: 13% faster, Airbnb: 8% faster) but slightly increases runtime for others (Bank: 33% slower). Importantly, consistency remains virtually unchanged ($\pm 0.001$), confirming that stability improvements do not compromise accuracy.

Table 6: **Local explanation performance.** Running times (seconds) for computing dataset-level SV using convergence criterion from equation 11 ($\epsilon = 0.025$). Consistency measured as Pearson correlation with KernelSHAP baseline. Bold indicates best running time among SIM-Shapley variants.

| Data | Model | Avg. Running Time (s) | | | Avg. Consistency | |
|---|---|---|---|---|---|---|
| | | SIM-Shapley | Stable-SIM-Shapley | KernelSHAP | SIM-Shapley | Stable-SIM-Shapley |
| Bike | XGBoost | **8.48** | 8.59 | 12.62 | 0.9594 | 0.9594 |
| Credit | CatBoost | 0.52 | **0.45** | 3.11 | 0.9831 | 0.9829 |
| Bank | CatBoost | **1.73** | 2.30 | 7.49 | 0.9816 | 0.9816 |
| Airbnb | MLP | 1.76 | **1.62** | 3.20 | 0.9111 | 0.9121 |

We also evaluate convergence behavior across all experimental settings (detailed in Appendix E.4). Stable-SIM-Shapley consistently exhibits smoother convergence trajectories with reduced oscillations compared to standard SIM-Shapley, while both variants achieve faster convergence than baselines without sacrificing accuracy.

## 5 DISCUSSION

Re-framing SV estimation as stochastic optimization—combining mini-batch sampling, momentum updates, and $\ell_2$ regularization—yields substantial speed and stability gains. SIM-Shapley converges linearly to the true solution (Theorem 2), with provable variance reduction at each step (Theorem 1), and registers negligible bias (Section 2.2). Empirically, on standard ML benchmarks and the MIMIC-IV-ED clinical data (Appendix E.5), SIM-Shapley and its Stable variant reduce computing time by up to 85% while producing explanations that align with domain knowledge.

SIM-Shapley preserves the weighted least-squares formulation, therefore all existing KernelSHAP enhancements—unbiased $A$ estimators, paired-sampling schemes, multiple imputation strategies and alternative cooperative-game definition—can be integrated seamlessly. The addition of an $\ell_2$ penalty strengthens convexity and invertibility, incurring no extra algorithmic complexity.

SIM-Shapley generalizes to any sampling-dependent quadratic minimization, providing a foundation for accelerating Monte Carlo approximations. Future work could explore adaptive momentum scaling, richer variance-reduction strategies, and extensions to other cooperative-game objectives beyond SV.

**Limitations and Future Work** The online algorithm in Section 2.2 does not provide an unbiased estimate of the true variance, since $\beta^n$ depends on previous iterations. We use it as a practical stopping heuristic that effectively tracks estimate stability, halting when fluctuations become small–validated by our experiments. Future improvements could use multiple mini-batches per iteration for more robust variance estimates. Additionally, while we focus on first-order SV (further discussed in Appendix A), extensions to higher-order Shapley interactions remain potential future work.

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

## A   RELATED WORK

In recent years, numerous Shapley Value (SV) based methods have been proposed in machine learning research, leveraging cooperative game theory to explain the contributions of different players.

For example, Ghorbani & Zou (2019); Jia et al. (2019); Wu et al. (2023) adopt SV to evaluate the contribution of each data instance, a framework commonly referred to as *Data Shapley*. In addition, SV has also been applied in ensemble learning, where the contribution of individual classifiers in a voting game is quantified (Rozemberczki & Sarkar, 2021).

Nevertheless, the players defined in the aforementioned methods differ substantially from our setting, where the players correspond to the features of a dataset. Specifically, data instances and classifiers in ensemble models can typically be regarded as i.i.d. players, making the value function for computing subset contributions relatively straightforward. In contrast, our focus on feature attribution requires accounting for the removal of features, which poses additional challenges.

**Imputation Strategy.**  Various strategies have been proposed to address feature removal when measuring feature importance using SV-based methods. Chen et al. (2022) categorized these strategies into *Conditional* (using the conditional distribution), *Marginal* (using the marginal distribution), *Baseline* (using a hybrid sample), and *None* (filling with zero). Although we adopt the marginal distribution by default for simplicity, our *imputer* module is designed with sufficient flexibility to accommodate other strategies, provided they are implemented correctly. Actually, we have used a surrogate model proposed in FastSHAP (Jethani et al., 2022) to obtain conditional distribution in baseline comparison.

**Model Agnostic.**  To improve computational efficiency, many approaches exploit the specific architecture of machine learning models. For instance, DeepSHAP (Lundberg & Lee, 2017), Deep ShapNet (Wang et al., 2021), and EmSHAP (Lu et al., 2024) are designed for neural networks, while TreeSHAP (Lundberg et al., 2020) is tailored to tree-based models. In stark contrast, our SIM-Shapley is a WLS-based method (see Section 1.1.1) for evaluating the Shapley Value of features in a *model-agnostic* setting. Notably, we focus on WLS-methods in this work, disregarding techniques like RegressionMSR (Witter et al., 2025) (with unavailable implementation) that combine Monte Carlo sampling with WLS, or rely solely on Monte Carlo. From this perspective, we compare SIM-Shapley against several contemporary SV methods with similar properties, in addition to KernelSHAP and SAGE:

- *FastSHAP* (Jethani et al., 2022): An amortized approach that trains a surrogate model to approximate the conditional distribution of features. The WLS estimator (5) is reparameterized as a multilayer perceptron to learn Shapley Values.

- *SimSHAP* (Zhang et al., 2024): An improved variant of FastSHAP. Its implementation, however, was not publicly available at the time of writing this paper.

- *Leverage SHAP* (Musco & Witter, 2025): A lightweight modification of KernelSHAP using leverage score sampling. Although flexible in principle, its implementation is limited. In practice, the method fixes the prediction game to model predictive performance and fills missing features with dataset mean values.

As illustrated above, both FastSHAP and SimSHAP rely on training new black-box models (e.g., MLPs) to explain an existing black-box model, which reduces transparency and is unacceptable in certain scenarios. Moreover, these approaches require designing and fine-tuning a neural network for each new task, introducing substantial overhead. In addition, all three methods are restricted to providing only local explanations for individual instances and are unsuitable for some supervised learning tasks.

**Prediction Game.**  Albeit the prediction-loss game has already been defined in Section 1.1.1, we also provide the definition of the prediction game here. This is to highlight the game-agnostic property of SIM-Shapley alongside KernelSHAP, and to ensure comparability with FastSHAP and LeverageSHAP, which are restricted to the prediction game:

$$v(S) = \mathbb{E}\big[f(x_S, X_{S^C})\big], \tag{14}$$

where $x_S$ denotes the observed subset of features, and $X_{S^C}$ is the corresponding random variable for the remaining features under the marginal distribution.

**Shapley Interaction.** In our setting, the Shapley Value fairly distributes the quantified contribution among individual players by evaluating all possible subsets. However, this perspective overlooks potential *synergies* or *redundancies* between entities (i.e., features in our context). To capture the predictive power of more complex machine learning models, researchers have proposed measuring the joint value of groups of entities, a concept known as *interaction* (Grabisch & Roubens, 1999; Fumagalli et al., 2023). Analogous to the idea of the SV, Shapley Interactions (SIs) (Bordt & von Luxburg, 2023; Grabisch & Roubens, 1999) allocate the overall worth to all groups of entities up to a maximum explanation order, with the standard SV corresponding to the first-order case.

Building on this idea, Muschalik et al. invent an open-source Python package, *shapiq* (Muschalik et al., 2024; Fumagalli et al., 2023; 2024), which unifies state-of-the-art algorithms for efficiently computing SVs and higher-order SIs in an application-agnostic framework. Within this package, KernelSHAP is incorporated and extended to approximate SIs. Since our proposed SIM-Shapley merely accelerates the estimation procedure of KernelSHAP without altering the definition of players, games, or values, it can be seamlessly integrated into the *shapiq* framework and is well-suited for complicated XAI scenarios. In addition, *shapiq* includes several novel approximators, such as ProxySPEX (Butler et al., 2025), a sparse Fourier-based method designed to efficiently capture higher-order interactions using a trained gradient boosting tree to model the hierarchical structure of LLM interactions.

## B  COMPLETE PROOFS

**Remark 1.** As $m \to \infty$, the empirical matrix $A$ converges in probability to a positive definite population matrix $\Sigma := \mathbb{E}[z_i z_i^\top]$, whose minimum eigenvalue has lower bound $\frac{1}{4}$.

*Proof.* Recall that the matrix $A$ is defined as:

$$A = \frac{1}{m} \sum_{i=1}^{m} z_i z_i^\top,$$

where each $z_i \in \{0, 1\}^d$ is drawn independently and uniformly from $\mathbb{R}^d$.

Let $\Sigma := \mathbb{E}[z_i z_i^\top]$ denote the population second-moment matrix. Since the $z_i$ are i.i.d. and bounded random vectors, applying the law of large numbers element-wise gives us:

$$A = \frac{1}{m} \sum_{i=1}^{m} z_i z_i^\top \xrightarrow{\text{a.s.}} \mathbb{E}[z_i z_i^\top] =: \Sigma.$$

Since $z_i$ is uniformly distributed over $\{0, 1\}^d$, its coordinates are independent and identically distributed Bernoulli($\frac{1}{2}$). Then

for diagonal entries ($k = \ell$):

$$\mathbb{E}[z_{ik}^2] = \mathbb{E}[z_{ik}] = \frac{1}{2};$$

for off-diagonal entries ($k \neq \ell$):

$$\mathbb{E}[z_{ik} z_{i\ell}] = \mathbb{E}[z_{ik}]\mathbb{E}[z_{i\ell}] = \frac{1}{4}.$$

Thus, the population matrix takes the explicit form:

$$\Sigma = \mathbb{E}[z_i z_i^\top] = \frac{1}{4}\mathbf{1}_d \mathbf{1}_d^\top + \frac{1}{4}I_d.$$

Multiplying a non-zero vector $v \in \mathbb{R}^d$ on the right hand side gives us:

$$\mathbb{E}[z_i z_i^T]v = \left(\frac{1}{4}\mathbf{1}_d \mathbf{1}_d^T + \frac{1}{4}I_d\right)v = \frac{1}{4}\mathbf{1}_d(\mathbf{1}_d^T v) + \frac{1}{4}v$$

As a result, its eigenvalues are:

- $\lambda_1 = \frac{1}{4}(d+1)$, corresponding to the direction $\mathbf{1}_d$,

- $\lambda_2 = \cdots = \lambda_d = \frac{1}{4}$, corresponding to any direction orthogonal to $\mathbf{1}_d$.

and $\alpha := \min(\lambda_1, \lambda_2) = \frac{1}{4}$. $\qquad\square$

### B.1 POOF OF THEOREM 1

*Proof.* Recall the estimator of KernelSHAP in Covert & Lee (2021) is defined as $\boldsymbol{\beta}$ and the recursive form of $\boldsymbol{\beta}^{(n+1)}$ is as following:

$$\boldsymbol{\beta}^{(n+1)} = (1-t) \sum_{j=1}^{n+1} t^{n+1-j} \boldsymbol{\delta}^{(j)}.$$

we see that each $\boldsymbol{\delta}^{(j)}$ depends on $\boldsymbol{\beta}^{(j-1)}$, making the iterations dependent through time.

We assume that the matrix $A = \frac{1}{m} \sum_{i=1}^{m} z_i z_i^T$ is positive definite with minimal eigenvalue $\alpha > 0$, which holds asymptotically as $m \to \infty$ by Remark 1. This guarantees that the regularized matrix $\bar{A} = A + \lambda I$ is strictly positive definite, and $\bar{A}^{-1}$ is well-conditioned and bounded. Consequently, the stochastic updates remain well-behaved and the propagation of variance across iterations can be controlled.

The variance propagates recursively as:

$$\mathrm{Var}(\boldsymbol{\beta}^{(n+1)}) = t^2 \mathrm{Var}(\boldsymbol{\beta}^{(n)}) + (1-t)^2 \mathrm{Var}(\boldsymbol{\delta}^{(n+1)}) + 2t(1-t) \mathrm{Cov}(\boldsymbol{\beta}^{(n)}, \boldsymbol{\delta}^{(n+1)}).$$

Assuming initialization $\boldsymbol{\beta}^{(0)} = \mathbf{0}$ and large $m$, the first step gives:

$$\mathrm{Var}(\boldsymbol{\beta}^{(1)}) = (1-t)^2 \mathrm{Var}(\boldsymbol{\delta}^{(1)}) \leq (1-t)^2 \mathrm{Var}(\boldsymbol{\beta}).$$

For subsequent steps, if $\mathrm{Var}(\boldsymbol{\delta}^{(j)}) \leq \mathrm{Var}(\boldsymbol{\beta})$ (which holds under $\ell_2$ regularization), and the cross-covariance remains bounded, then:

$$\mathrm{Var}(\boldsymbol{\beta}^{(2)}) = t^2(1-t)^2 \mathrm{Var}(\boldsymbol{\delta}^{(1)}) + (1-t)^2 \mathrm{Var}(\boldsymbol{\delta}^{(2)}) + 2t(1-t)^2 \mathrm{Cov}(\boldsymbol{\delta}^{(1)}, \boldsymbol{\delta}^{(2)})$$
$$\leq (1-t)^2 \mathrm{Var}(\boldsymbol{\delta}^{(1)}) \leq (1-t)^2 \mathrm{Var}(\boldsymbol{\beta}),$$

$$\vdots$$

$$\mathrm{Var}(\boldsymbol{\beta}^{(n+1)}) \leq (1-t)^2 \max_{j \leq n+1} \mathrm{Var}(\boldsymbol{\delta}^{(j)}) \leq (1-t)^2 \mathrm{Var}(\boldsymbol{\beta}).$$

We emphasize that $\boldsymbol{\delta}^{(n+1)}$ is computed conditional on $\boldsymbol{\beta}^{(n)}$, so independence does not hold. The covariance term $\mathrm{Cov}(\boldsymbol{\beta}^{(n)}, \boldsymbol{\delta}^{(n+1)})$ cannot be assumed to vanish, but is expected to remain bounded under regularized updates. By the Cauchy-Schwarz inequality:

$$|\mathrm{Cov}(\boldsymbol{\beta}^{(n)}, \boldsymbol{\delta}^{(n+1)})| \leq \sqrt{\mathrm{Var}(\boldsymbol{\beta}^{(n)}) \cdot \mathrm{Var}(\boldsymbol{\delta}^{(n+1)})},$$

thus the recursion remains stable if both variances are controlled.

In conclusion, the variance of the SIM-Shapley estimator $\boldsymbol{\beta}^{(n+1)}$ is at most of the same order as that of the KernelSHAP estimator $\boldsymbol{\beta}$, and under ideal conditions may even be strictly smaller. This confirms that the variance reduction effect of the $\ell_2$ regularization is preserved across iterations. $\quad\square$

### B.2 POOF OF THEOREM 2

*Proof.* Under the assumption that $m \to \infty$ (Remark 1), we first prove the existence of optimal solution of SIM-Shapley, then analyze its quotient convergence rate,

### B.2.1 EXISTENCE OF A FIXED POINT

Given a vector $\boldsymbol{\beta}$, define $\delta^*(\boldsymbol{\beta})$ as the unique solution to the *population-level* version of problem equation 7, where all empirical averages are replaced by their exact expectations. Define the mapping:

$$T : \mathbb{R}^d \to \mathbb{R}^d, \quad T(\boldsymbol{\beta}) = t\,\boldsymbol{\beta} + (1-t)\,\delta^*(\boldsymbol{\beta}).$$

A vector $\boldsymbol{\beta}^*$ is a fixed point if

$$\boldsymbol{\beta}^* = T(\boldsymbol{\beta}^*) = t\,\boldsymbol{\beta}^* + (1-t)\,\delta^*(\boldsymbol{\beta}^*). \tag{15}$$

This implies directly that

$$\delta^*(\boldsymbol{\beta}^*) = \boldsymbol{\beta}^*,$$

i.e., the fixed point $\boldsymbol{\beta}^*$ satisfies the optimality condition of the constrained quadratic program.

Since the objective function in equation 7 is strictly convex and smooth, and the constraint is affine, the solution mapping $\delta^*(\boldsymbol{\beta})$ is continuous in $\boldsymbol{\beta}$, implying that $T$ is continuous. Moreover, due to the quadratic growth of the objective, there exists a compact, convex set $\mathcal{K} \subset \mathbb{R}^d$ (e.g., a sufficiently large closed ball) such that $T(\mathcal{K}) \subset \mathcal{K}$.

Applying Brouwer's fixed-point theorem to the continuous self-map $T : \mathcal{K} \to \mathcal{K}$, we conclude that a fixed point $\boldsymbol{\beta}^* \in \mathcal{K}$ exists.

### B.2.2 LOCAL LINEARIZATION AND CONVERGENCE RATE VIA THE JACOBIAN

To analyze the convergence behavior near the fixed point $\boldsymbol{\beta}^*$, we linearize the mapping $T$ in a neighborhood of $\boldsymbol{\beta}^*$.

Let the error at iteration $n$ be:
$$e^{(n)} = \boldsymbol{\beta}^{(n)} - \boldsymbol{\beta}^*.$$

Since $\boldsymbol{\beta}^*$ is a fixed point, $T(\boldsymbol{\beta}^*) = \boldsymbol{\beta}^*$. Using the differentiability of $\delta^*(\boldsymbol{\beta})$, we apply a first-order Taylor expansion around $\boldsymbol{\beta}^*$:

$$\delta^*(\boldsymbol{\beta}) \approx \delta^*(\boldsymbol{\beta}^*) + M(\boldsymbol{\beta} - \boldsymbol{\beta}^*),$$

where

$$M = \left.\frac{\partial \delta^*(\boldsymbol{\beta})}{\partial \boldsymbol{\beta}}\right|_{\boldsymbol{\beta}=\boldsymbol{\beta}^*}$$

is the Jacobian evaluated at $\boldsymbol{\beta}^*$. Since $\delta^*(\boldsymbol{\beta}^*) = \boldsymbol{\beta}^*$, this simplifies to:

$$\delta^*(\boldsymbol{\beta}) \approx \boldsymbol{\beta}^* + M(\boldsymbol{\beta} - \boldsymbol{\beta}^*).$$

Substituting into the update rule:

$$\boldsymbol{\beta}^{(n+1)} = t\,\boldsymbol{\beta}^{(n)} + (1-t)\,\delta^*(\boldsymbol{\beta}^{(n)}),$$

we obtain:

$$\begin{aligned}
\boldsymbol{\beta}^{(n+1)} &\approx t\,\boldsymbol{\beta}^{(n)} + (1-t)\left[\boldsymbol{\beta}^* + M(\boldsymbol{\beta}^{(n)} - \boldsymbol{\beta}^*)\right] \\
&= t\,\boldsymbol{\beta}^{(n)} + (1-t)\boldsymbol{\beta}^* + (1-t)M(\boldsymbol{\beta}^{(n)} - \boldsymbol{\beta}^*).
\end{aligned}$$

Subtracting $\boldsymbol{\beta}^*$ from both sides yields:

$$e^{(n+1)} \approx \left[tI + (1-t)M\right] e^{(n)}.$$

Define the linear update operator $G := tI + (1-t)M$. Then, the convergence behavior is governed by the spectral radius:

$$\rho(G) = \max\{|\lambda| : \lambda \in \sigma(G)\}.$$

If $\rho(G) < 1$, the error converges linearly:

$$\|e^{(n)}\| = O(\rho(G)^n).$$

In our setting, the matrix $A$ is given by:

$$A = \frac{1}{m} \sum_{i=1}^{m} z_i z_i^T,$$

and the regularized version is $\bar{A} = A + \lambda I$. Through differentiation of the closed-form expression for $\delta^*(\boldsymbol{\beta})$ under the equality constraint, we obtain the Jacobian:

$$M = -\frac{t}{1-t} \bar{A}^{-1} \left[ A - \mathbf{1} \cdot \frac{\mathbf{1}^T (\bar{A}^{-1} A - I)}{\mathbf{1}^T \bar{A}^{-1} \mathbf{1}} \right].$$

Thus, the linear operator $G$ becomes:

$$G = tI + (1-t)M = t \left[ I - \bar{A}^{-1} A + \bar{A}^{-1} \mathbf{1} \cdot \frac{\mathbf{1}^T (\bar{A}^{-1} A - I)}{\mathbf{1}^T \bar{A}^{-1} \mathbf{1}} \right].$$

In the subspace orthogonal to the constraint direction, assume $x$ is an eigenvector of $\bar{A}^{-1} A$ with eigenvalue $\mu = \frac{\alpha_i}{\alpha_i + \lambda} < 1$, where $\alpha_i$ is an eigenvalue of $A$. Then,

$$Gx \approx t(1 - \mu)x.$$

Hence, the spectral radius of $G$ is:

$$\rho(G) = \max_i |t(1 - \mu_i)| = t \cdot \frac{\lambda}{\alpha_{\min}^+ + \lambda},$$

where $\alpha_{\min}^+$ denotes the smallest nonzero eigenvalue of $A$, which has a lower bound (Remark 1). Since $0 < t < 1$ and $\frac{\lambda}{\alpha_i + \lambda} < 1$, we conclude $\rho(G) < 1$, and the iteration converges linearly:

$$\|e^{(n)}\| = O(\rho(G)^n).$$

$\square$

## C    SIM-SHAPLEY: GLOBAL EXPLANATION VERSION

We exhibit SIM-Shapley for global explanation in Algorithm 2.

---
**Algorithm 2 SIM-Shapley** (Global Explanation)

---
**Input:** Global reference set $(X_{\text{REF}}, Y_{\text{REF}})$ as $p(U)$, background set $(X_{\text{BK}}, Y_{\text{BK}})$, subset sample size $m$, batch size $B$, number of iterations $T$
**Output:** Final estimate $\boldsymbol{\beta}^{(n)}$
1: Initialize $\boldsymbol{\beta}^{(0)} = \mathbf{0}$; $Mean_0 = \mathbf{0}$; $S_0 = \mathbf{0}$; precompute $V(\mathbf{1})$ and $V(\mathbf{0})$ by averaging over $(X_{\text{REF}}, Y_{\text{REF}})$
2: **for** $n = 1$ to $T$ **do**
3:     Independently sample $\{z_j\}_{j=1}^{m} \sim p(z)$
4:     Sample $\{(x_j, y_j)\}_{j=1}^{B} \sim p(U)$
5:     For each $z_j$, estimate $V(z_j)$ by averaging predictions on $\{(x_j, y_j)\}_{j=1}^{B}$, marginalizing missing features using the background set
6:     Solve equation 7 to obtain $\boldsymbol{\delta}^{(n)}$ and update $\boldsymbol{\beta}^{(n)}$
7:     Update running variance $Var_n$ of $\boldsymbol{\beta}^{(n)}$ using Welford's algorithm:
8:       $\Delta \leftarrow \boldsymbol{\beta}^{(n)} - Mean_{n-1}$; $Mean_n \leftarrow Mean_{n-1} + \Delta/n$
9:       $S_n \leftarrow S_{n-1} + \Delta \cdot (\boldsymbol{\beta}^{(n)} - Mean_n)$; $Var_n \leftarrow S_n/(n-1)$
10:    **if** convergence criterion satisfied **then**
11:      **break**
12:    **end if**
13: **end for**
14: **return** $\boldsymbol{\beta}^{(n)}$

---

---

**Algorithm 3 Stable SIM-Shapley** (Local Explanation)

---

**Input:** Instance $(x, y)$, background set $(X_{\text{BK}}, Y_{\text{BK}})$, subset sample size $m$, number of iterations $T$
**Output:** Final estimate $\boldsymbol{\beta}^{(n)}$
1: Initialize $\boldsymbol{\beta}^{(0)} = \mathbf{0}$; $Mean_0 = \mathbf{0}$; $S_0 = \mathbf{0}$; precompute $v(\mathbf{1})$ and $v(\mathbf{0})$ for the target instance
2: **for** $n = 1$ to $T$ **do**
3:    Independently sample $\{z_j\}_{j=1}^m \sim p(z)$
4:    For each $z_j$, compute $v(z_j)$ by marginalizing missing features using the background set
5:    Solve equation 16 to obtain $\boldsymbol{\delta}^{(n)}$ and update $\boldsymbol{\beta}^{(n)}$
6:    Update running variance $Var_n$ of $\boldsymbol{\beta}^{(n)}$ using Welford's algorithm:
7:       $\Delta \leftarrow \boldsymbol{\beta}^{(n)} - Mean_{n-1}$; $Mean_n \leftarrow Mean_{n-1} + \Delta/n$
8:       $S_n \leftarrow S_{n-1} + \Delta \cdot (\boldsymbol{\beta}^{(n)} - Mean_n)$; $Var_n \leftarrow S_n/(n-1)$
9:    **if** convergence criterion satisfied **then**
10:       **break**
11:    **end if**
12:    **if** negative sampling detected **then**
13:       Let $\boldsymbol{\delta}^{(n)} = \boldsymbol{\delta}^{(n-1)}, \boldsymbol{\beta}^{(n)} = \boldsymbol{\beta}^{(n-1)}$
14:    **end if**
15: **end for**
16: **return** $\boldsymbol{\beta}^{(n)}$

---

## D  STABLE SIM-SHAPLEY

In this part, we introduce Algorithm 3 and 4 that is equipped with stability improving methods proposed in our paper. New iteration form and explicit solution of $\boldsymbol{\delta}$ are as following:

**New iteration form with correcting initial deviation:**

$$
\min_{\beta_0^{(n+1)},\ldots,\beta_d^{(n+1)}} \frac{1}{m} \sum_{i=1}^m \left( v(\mathbf{0}) + z_i^T \boldsymbol{\beta}^{(n+1)} - v(z_i) \right)^2 + \lambda \left\| \boldsymbol{\delta}^{(n+1)} \right\|_2^2
$$

$$
\text{s.t.} \quad \mathbf{1}^T \boldsymbol{\beta}^{(n+1)} = v(\mathbf{1}) - v(\mathbf{0}), \quad \boldsymbol{\beta}^{(n+1)} = \frac{t\boldsymbol{\beta}^n}{1 - t^{n+1}} + \frac{(1-t)\boldsymbol{\delta}^{(n+1)}}{1 - t^{n+1}}
\tag{16}
$$

**New explicit solution with correcting initial deviation:**

$$
c_1 = \frac{t}{1 - t^{n+1}}, \quad c_2 = \frac{1 - t}{1 - t^{n+1}}
$$

$$
\boldsymbol{\delta}^{(n+1)} = \frac{\bar{A}^{-1}[(\bar{b} - c_1 A \boldsymbol{\beta}^{(n)}) + \mathbf{1} \frac{(v(\mathbf{1}) - v(\mathbf{0})) - c_1 \mathbf{1}^T \boldsymbol{\beta}^{(n)} - \mathbf{1}^T \bar{A}^{-1}(\bar{b} - c_1 A \boldsymbol{\beta}^{(n)})}{\mathbf{1}^T \bar{A}^{-1} \mathbf{1}}]}{c_2}.
\tag{17}
$$

## E  ADDITIONAL EXPERIMENTAL DETAILS

For fair comparison, all methods approximate the conditional distribution $p(X_{\bar{S}} \mid X_S = x_S)$ using marginal distributions estimated from the background dataset (see Section 2.1). Mean squared error (MSE) is used for regression tasks (i.e., Airbnb and Bike) and cross-entropy (CE) is used for classifications (i.e., Bank and Credit). For each experiment on the same dataset and explanation, we report the average wall-clock time under a unified convergence threshold.

### E.1  PEARSON'S CORRELATION COEFFICIENT AS A CONSISTENCY METRIC

In the main paper, we introduced several methods for approximating Shapley values. Under a common cooperative game formulation, these estimates should exhibit consistency, even though individual values may vary due to randomness in the sampling process. That is, while the exact numerical outputs may differ, the overall distribution of estimated Shapley values is expected to remain stable across methods.

---

**Algorithm 4 Stable SIM-Shapley** (Global Explanation)

---

**Input:** Global reference set $(X_{\text{REF}}, Y_{\text{REF}})$ as $p(U)$, background set $(X_{\text{BK}}, Y_{\text{BK}})$, subset sample size $m$, batch size $B$, number of iterations $T$
**Output:** Final estimate $\boldsymbol{\beta}^{(n)}$
1: Initialize $\boldsymbol{\beta}^{(0)} = \mathbf{0}$; $Mean_0 = \mathbf{0}$; $S_0 = \mathbf{0}$; precompute $V(\mathbf{1})$ and $V(\mathbf{0})$ by averaging over $(X_{\text{REF}}, Y_{\text{REF}})$
2: **for** $n = 1$ to $T$ **do**
3:      Independently sample $\{z_j\}_{j=1}^{m} \sim p(z)$
4:      Sample $\{(x_j, y_j)\}_{j=1}^{B} \sim p(U)$
5:      For each $z_j$, estimate $V(z_j)$ by averaging predictions on $\{(x_j, y_j)\}_{j=1}^{B}$, marginalizing missing features using the background set
6:      Solve equation 16 to obtain $\boldsymbol{\delta}^{(n)}$ and update $\boldsymbol{\beta}^{(n)}$
7:      Update running variance $Var_n$ of $\boldsymbol{\beta}^{(n)}$ using Welford's algorithm:
8:        $\Delta \leftarrow \boldsymbol{\beta}^{(n)} - Mean_{n-1}$; $\ Mean_n \leftarrow Mean_{n-1} + \Delta/n$
9:        $S_n \leftarrow S_{n-1} + \Delta \cdot (\boldsymbol{\beta}^{(n)} - Mean_n)$; $\ Var_n \leftarrow S_n/(n-1)$
10:     **if** convergence criterion satisfied **then**
11:        **break**
12:     **end if**
13:     **if** negative sampling detected **then**
14:        Let $\boldsymbol{\delta}^{(n)} = \boldsymbol{\delta}^{(n-1)}, \boldsymbol{\beta}^{(n)} = \boldsymbol{\beta}^{(n-1)}$
15:     **end if**
16: **end for**
17: **return** $\boldsymbol{\beta}^{(n)}$

---

To evaluate this consistency, prior work has employed regression analysis. As illustrated in Figure 1, the estimates from different methods tend to align closely along a straight line, indicating a strong linear relationship. This observation motivates the use of *Pearson's correlation coefficient* as a quantitative measure of consistency. A coefficient near $1$ suggests near-perfect linear agreement between two sets of Shapley estimates.

Moreover, when using the same cooperative game and an identical strategy for approximating the conditional distribution of missing features, the imposed equality constraint ensures that any bias remains bounded.

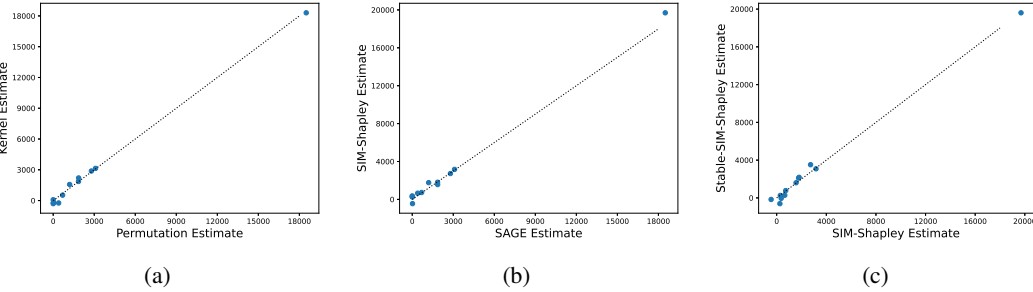

(a)                    (b)                    (c)

eFigure 1: **Comparison of consistency across different Shapley value estimators.** All axes represent estimated Shapley values. (a) Comparison between two SAGE estimates obtained via permutation and kernel regression. (b) Comparison between the SAGE estimate and the SIM-Shapley estimate. Experiments are conducted on the Bank dataset under identical settings. (c) Comparison between SIM-Shapley and its stable version. All SV estimates stem from Bike dataset with global explanation.

### E.2 DATASETS

We exploit following datasets: the Bike Sharing Demand dataset (Fanaee-T & Gama, 2013), the South German Credit dataset (sou, 2019), the Portuguese Bank Marketing dataset (Moro & Cortez,

2014), the American 1994 Census dataset (Becker & Kohavi, 1996), the NYC Airbnb Listings dataset (Gomonov, 2019), the MIMIC-IV Emergency Department (MIMIC-IV-ED) dataset (Johnson et al., 2023) and The MNIST database of handwritten digits [2].

Without recording in the main paper, the predictive model for Census dataset is LightGBM (Shi et al., 2025) and MNIST is predicted by a simple CNN implemented by PyTorch.

We provide supplementary details for the experimental setup described in the main paper. Each dataset is split into training, validation, and test sets with a ratio of 7:2:1. Table 1 summarizes the dataset statistics and experimental configurations.

### E.3 EMPIRICAL EXPERIMENTS FOR HYPER-PARAMETERS

equation 11 and equation 12 introduce the hyper-parameters $\epsilon$ and $\xi$. To empirically demonstrate their impact on the algorithm's performance, we designed experiments that systematically vary their values and report the resulting bias or convergence ratio.

Figure 2 illustrates the relationship between the convergence ratio and the bias against the ground truth. This naturally indicates that as the algorithm achieves closer proximity to the ground-truth value, more iterations are required for convergence. Correspondingly, Figure 3 demonstrates that the choice of $\{\mathbf{0.30}, \mathbf{0.35}\}$ discussed in Section 3 yields the most efficient reduction in convergence time.

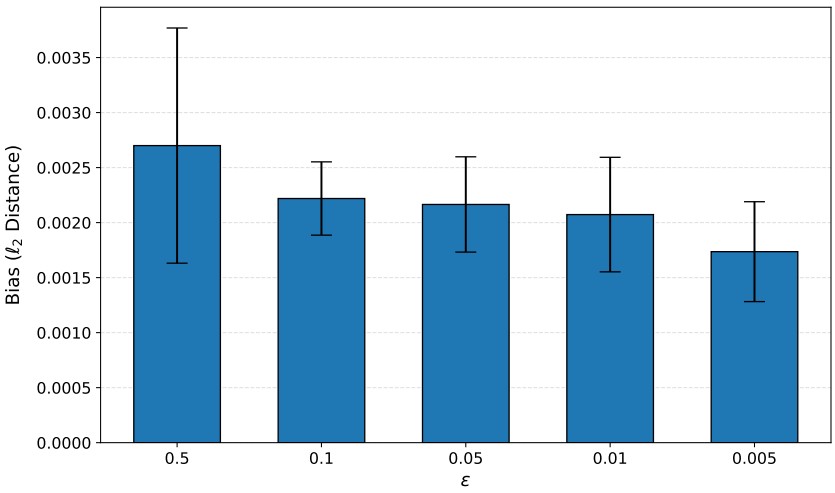

eFigure 2: Bias-Convergence Ratio ($\epsilon$ in equation 11) relationship for SIM-Shapley on the Bank dataset. Bias represents the $\ell_2$ distance from ground-truth SV as $\epsilon$ varies.

### E.4 ML TASKS DETAILS

A portion of the training set is used as the background set. For global explanations, we sample a subset of the test set—either $10\%$ or $5\%$ of the full dataset—as the global reference set to estimate the expectation of the cooperative game $V(S)$. For large-scale datasets such as MIMIC-IV, a smaller subset is used for efficiency. For local explanations, a single test instance is selected. The mini-batch size for sampling $z_i$ is set to 10 times the feature dimension, and the global batch size for data points is fixed at $512$.

The convergence threshold is kept consistent across all Shapley value–based methods to ensure meaningful comparisons in terms of consistency (i.e., Pearson correlation coefficient close to 1). We fix the regularization parameter at $\lambda = 0.01$, and set the EMA coefficient $t$ to either 0.5 or 0.55, as shown in Table 1.

---

[2] http://yann.lecun.com/exdb/mnist/

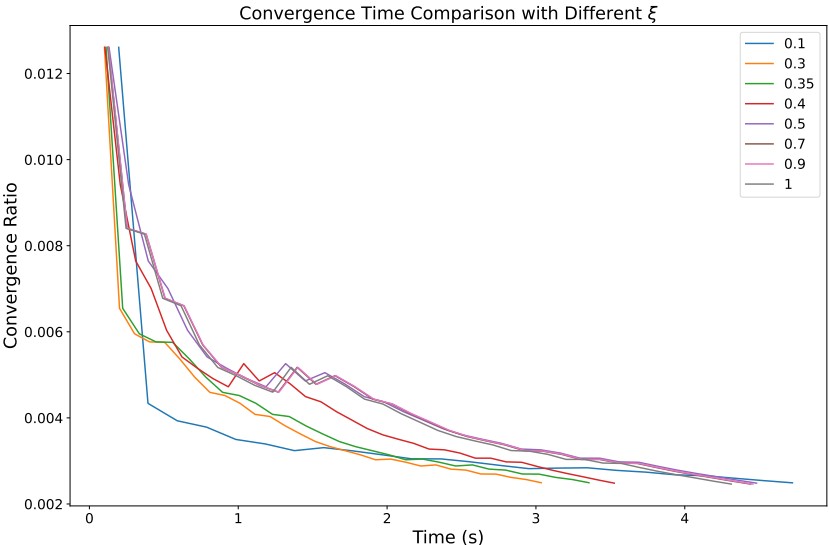

eFigure 3: Comparison of convergence time(s) under different threshold of negative sampling ($\xi$ in equation 12) for SIM-Shapley on the Bank dataset.

eTable 1: **Dataset profile and experiment settings**. $t$: the hyperparameter in update formula, which may be different in global and local explanation. Shapley Value estimation tasks on Census and MNIST only use local explanation with surrogate imputation; therefore, specific terms are omitted.

| Dataset | Total Size | Feature Dimension | Background Set Size | Global Reference Set Size | z Sample Size | Batch Size | t (Global/Local) |
|---|---|---|---|---|---|---|---|
| Bike | 10886 | 12 | 512 | 1088 | 120 | 512 | 0.55 / 0.50 |
| Credit | 1000 | 20 | 256 | 100 | 200 | 512 | 0.55 / 0.55 |
| Bank | 45210 | 16 | 512 | 2000 | 160 | 512 | 0.55 / 0.50 |
| Airbnb | 48590 | 15 | 1024 | 4859 | 150 | 512 | 0.55 / 0.50 |
| MIMIC-IV | 414537 | 56 | 2048 | 10000 | 560 | 512 | 0.55 / 0.55 |
| Census | 45222 | 14 | - | - | 512 | 512 | - / 0.55 |
| MNIST | 70000 | 784 | - | - | 512 | 512 | - / 0.50 |

For benchmarks, we compare against SAGE algorithm proposed in Covert et al. (2020) for global explanations, and original KernelSHAP (Lundberg & Lee, 2017) for local explanations, excluding the enhancements proposed in Covert & Lee (2021), which remain applicable to SIM-Shapley.

The MLP architectures used in our experiments are summarized in Table 2 and implemented in PyTorch. Each model contains three linear layers and two ELU activations, with varying widths across datasets. A Sigmoid activation is used for binary classification on the MIMIC-IV dataset. Both models are trained with a learning rate of 0.001 for up to 250 epochs using early stopping based on validation loss. The complete training pipeline is available in our open-source repository.

For baselines, we employ the same surrogate models and explainer models in FastSHAP (Jethani et al., 2022). To reduce half parameters in its explainer, we shrink the width of neural networks and remove one hidden layer. Meanwhile, we fill the removed features with the mean value of the whole dataset for SIM-Shapley and KernelSHAP to keep consistent with the imputation strategy of leverageSHAP.

Figures 4 and 5 summarize the consistency and runtime statistics from the independently conducted experiments reported in the main paper. Figure 6 demonstrates the accelerated convergence trend of SIM-Shapley and Stable-SIM-Shapley, providing further support for the advantage of our method.

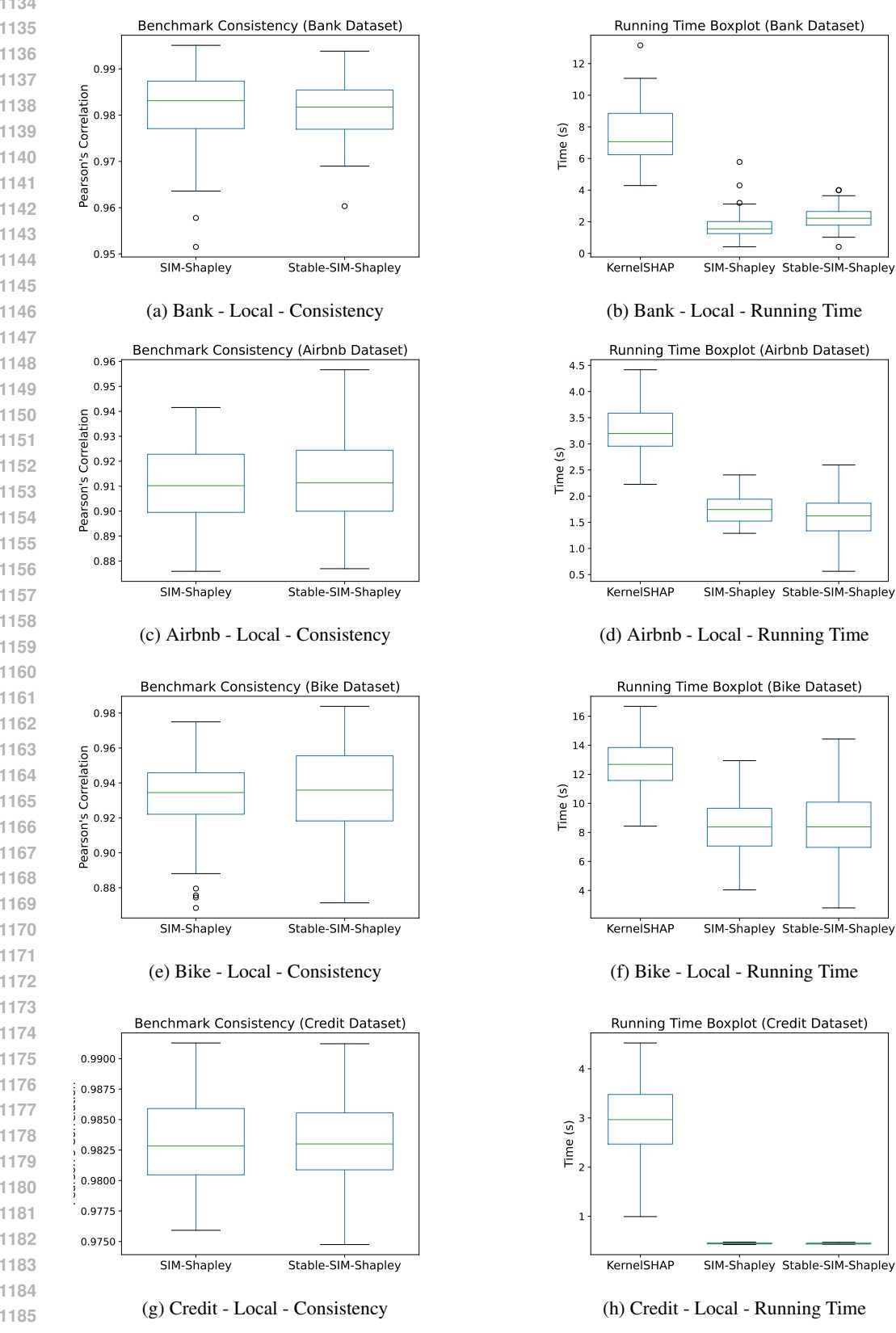

eFigure 4: **Boxplot of local explanation.** Comparison of Consistency (Pearson's Correlation) and Running Time(s).

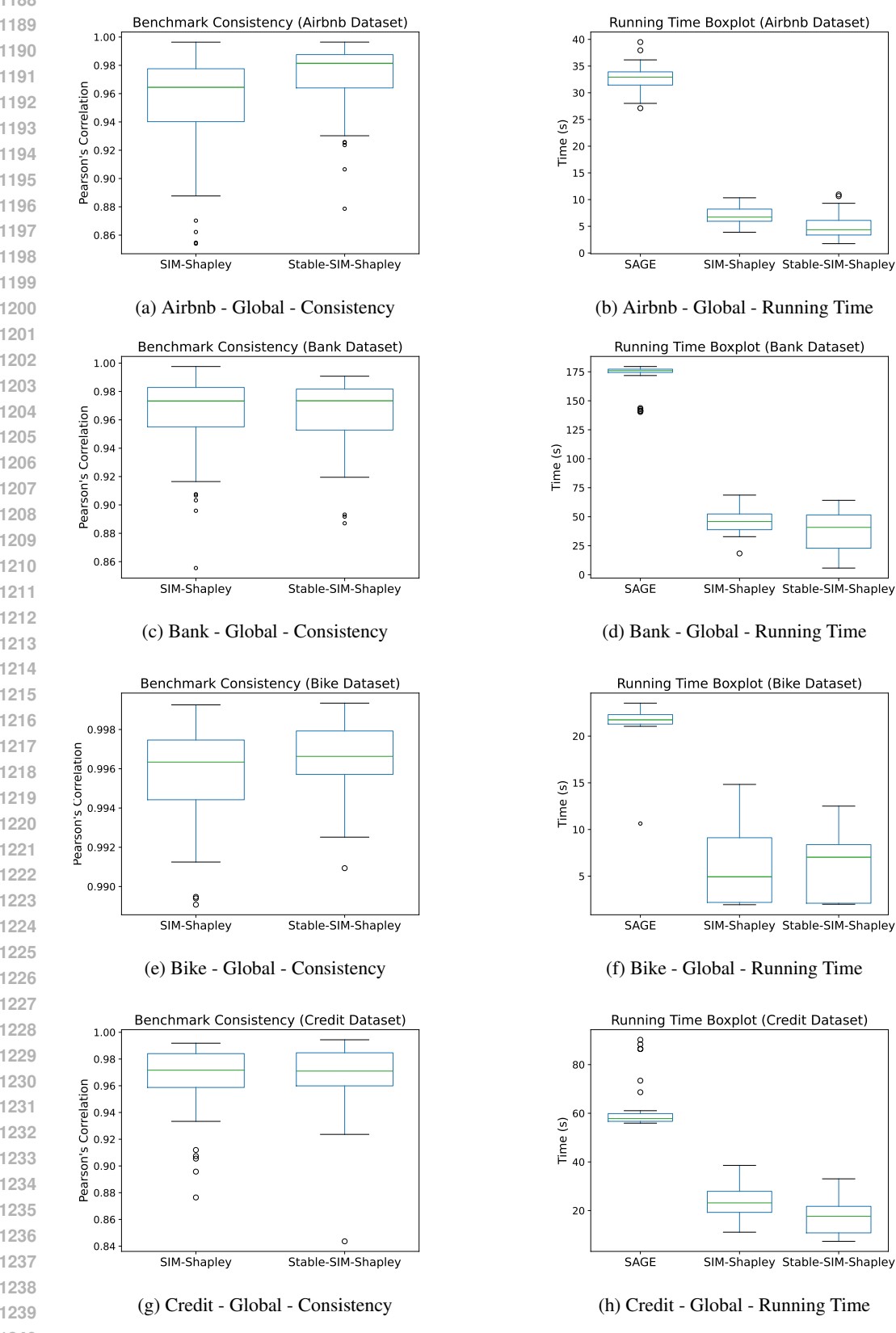

(a) Airbnb - Global - Consistency

(b) Airbnb - Global - Running Time

(c) Bank - Global - Consistency

(d) Bank - Global - Running Time

(e) Bike - Global - Consistency

(f) Bike - Global - Running Time

(g) Credit - Global - Consistency

(h) Credit - Global - Running Time

eFigure 5: **Boxplot of global explanation.** Comparison of Consistency (Pearson's Correlation) and Running Time(s).

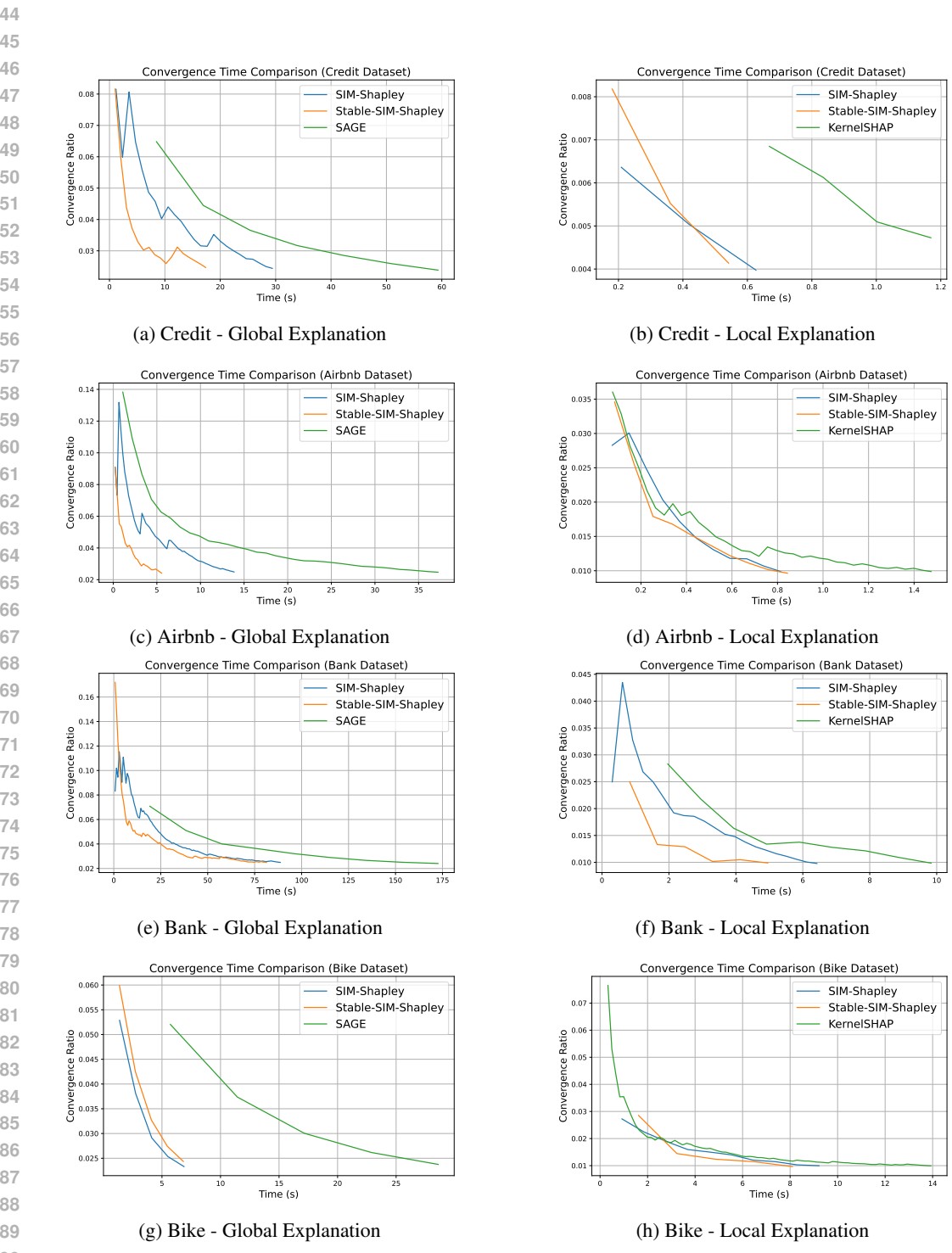

eFigure 6: **Comparison of convergence time(s).** Compare global and local interpretability methods on standard machine learning datasets.

eTable 2: **Comparison of two neural network architectures.**

| Layer # | MLP for Airbnb | MLP for MIMIC-IV |
|---------|----------------|------------------|
| 1 | Linear(15, 512) | Linear(56, 512) |
| 2 | ELU | ELU |
| 3 | Linear(512, 512) | Linear(512, 256) |
| 4 | ELU | ELU |
| 5 | Linear(512, 1) | Linear(256, 1) |
| 6 | — | Sigmoid |

eTable 3: **Average bias ($\ell_2$ distance) and runtime (seconds $\times 10^3$) across sample sizes on Credit dataset.** Times for SIM-Shapley and KernelSHAP correspond to estimation time, whereas FastSHAP entries report the neural network training time. FastSHAP* denotes the reduced-parameter version.

| Method | Metric | Sample Size | | | | | | | | | |
|--------|--------|------|------|------|------|------|------|------|------|------|------|
| | | 64 | 256 | 512 | 768 | 1024 | 1536 | 2048 | 2304 | 2816 | 3840 |
| SIM-Shapley | Bias | 0.028 | 0.028 | 0.019 | 0.019 | 0.019 | 0.018 | 0.018 | 0.018 | 0.018 | 0.018 |
| | Time | 6.92 | 5.81 | 5.97 | 10.64 | 12.27 | 14.70 | 19.88 | 26.44 | 33.82 | 42.26 |
| KernelSHAP | Bias | 0.012 | 0.012 | 0.012 | 0.009 | 0.008 | 0.007 | 0.006 | 0.006 | 0.005 | 0.004 |
| | Time | 8.36 | 11.66 | 9.87 | 15.05 | 15.37 | 23.83 | 30.65 | 32.99 | 34.56 | 54.91 |
| FastSHAP | Bias | 0.12 | | | | | | | | | |
| | Time | 12800 | | | | | | | | | |
| FastSHAP* | Bias | 0.13 | | | | | | | | | |
| | Time | 12190 | | | | | | | | | |

### E.5 REAL-WORLD INTERPRETABILITY ASSESSMENT

To validate our methods in a realistic clinical setting, we apply SIM-Shapley and Stable-SIM-Shapley to the MIMIC-IV-ED dataset (Johnson et al., 2023), focusing on in-hospital mortality prediction (see Appendix E.5.1 for dataset details). In addition to quantitative results (Section 4.3), we incorporate expert review from a board-certified emergency physician, who confirms that the top-ranked features (provided in Table 4)–such as vital signs, age, and comorbidity burden–are aligned with established medical knowledge. This supports the interpretability and practical utility of our methods. Full commentary appears in Appendix E.5.2.

#### E.5.1 COHORT FORMATION

The MIMIC-IV Emergency Department (MIMIC-IV-ED) dataset (Johnson et al., 2023) is a publicly available resource containing over 400,000 emergency department (ED) visit episodes. Following the data extraction pipeline proposed by Xie et al. (2022), we constructed a master dataset and formed our study cohort by excluding encounters with missing values in any of the 56 commonly used ED variables–such as demographics, vital signs, and comorbidities–selected for analysis.

#### E.5.2 DOMAIN-GUIDED INTERPRETATION OF RESULTS

To assess the clinical relevance of the feature attributions generated by SIM-Shapley and Stable-SIM-Shapley on the MIMIC-IV-ED dataset, we consulted a board-certified emergency physician. The physician independently reviewed the top-ranked features identified by our methods for predicting inpatient mortality and confirmed that they are plausible and aligned with clinical understanding. Notably, features such as vital signs, age, and comorbidity burden were consistent with known risk factors in emergency medicine. This alignment with expert knowledge supports the interpretability

and practical utility of our proposed method in real-world clinical settings. These observations align with domain standards such as the Canadian Triage and Acuity Scale (CTAS)[3], and comorbidity scoring systems like the Charlson Comorbidity Index (Schuttevaer et al., 2022; Murray et al., 2006).

---

[3]https://files.ontario.ca/moh_3/moh-manuals-prehospital-ctas-paramedic-guide-v2-0-en-2016-1
pdf

eTable 4: **Top-10 features identified by each SV method on the MIMIC cohort.** A check mark (✓) indicates inclusion in the method's top-10 ranking. Results are shown for both global and local explanations.

| Feature | Global Explanation | | | Local Explanation | | |
|---|---|---|---|---|---|---|
| | SAGE | SIM-Shapley | Stable-SIM | KernelSHAP | SIM-Shapley | Stable-SIM |
| Abdominal Pain | | ✓ | | | | |
| AIDS | | ✓ | | | | |
| Age | ✓ | ✓ | ✓ | ✓ | ✓ | ✓ |
| Congestive heart failure | | | | | ✓ | ✓ |
| Diabetes with complications | | | | | ✓ | ✓ |
| Diastolic blood pressure | ✓ | ✓ | | | | |
| Drug abuse | | ✓ | | | | |
| Fever/chills | | | ✓ | | | |
| Fluid and electroyte disorders | | | ✓ | | | |
| Gender | | | | ✓ | ✓ | ✓ |
| Heart rate at triage | ✓ | | ✓ | ✓ | ✓ | ✓ |
| Hemiplegia | ✓ | | | | | |
| Hypertension–complicated | | ✓ | ✓ | ✓ | ✓ | ✓ |
| Hypertension–uncomplicated | | | | | ✓ | |
| Local tumor, leukemia and lymphoma | | ✓ | | | | |
| Metastatic solid tumor | ✓ | ✓ | | | | |
| Myocardial infarction | | | ✓ | | | |
| Number of ED admission within the past year | | | | ✓ | | ✓ |
| Number of hospitalizations within the past year | | | | ✓ | ✓ | |
| Obesity | | | | ✓ | | |
| Peripheral oxygen saturation | ✓ | | ✓ | ✓ | | |
| Peripheral vascular disease | | ✓ | | | | ✓ |
| Psychoses | ✓ | | | | | |
| Respiration rate at triage | ✓ | | ✓ | | | ✓ |
| Stroke | ✓ | | | | | |
| Systolic blood pressure | ✓ | | | ✓ | | |
| Syncope | | ✓ | | | | |
| Temperature at triage | | | | ✓ | ✓ | ✓ |
| Weight Loss | | | ✓ | | | |

## F  LLM Usage Declaration

We used Claude (Anthropic) as a writing-assistance tool to improve grammar and clarity during manuscript preparation. All research ideas, designs, and analyses were conducted by the authors, who take full responsibility for the accuracy and integrity of the content.

