# OpenReview forum: "SIM-Shapley: A Stable and Computationally Efficient Approach to Shapley Value Approximation"
_ICLR.cc/2026/Conference — Submitted to ICLR 2026_

### Official Review · Reviewer_ZozL · 2025-10-27

**Soundness:** 4
**Presentation:** 3
**Contribution:** 3
**Rating:** 4
**Confidence:** 5

**Summary:**

The paper presents SIM-Shapley (SIM-SHAP) as a novel approximation method to estimate Shapely values. SIM-SHAP is quite related to many amortization and approximate methods relying on the weighted least squares representation of the Shapley value. SIM-SHAP extends the well known KernelSHAP approxmation method and Unbiased KernelSHAP (Covert and Lee, 2021) with an exponential moving average throughout the sampling and Shapley value computation procedure. The paper concludes by evaluating the approximation quality of SIM-SHAP compared to state-of-the-art methods. All in all the empirical evaluation shows that SIM-SHAP does not set the new state of the art but is comparable to it.

The following contains the **references** used throughout the review:
- [1] https://inria.hal.science/hal-03414720v1/document
- [2] https://arxiv.org/pdf/2506.11849
- [3] https://ojs.aaai.org/index.php/AAAI/article/view/29225
- [4] https://www.sciencedirect.com/science/article/abs/pii/S0305054808000804

**Strengths:**

- **Important Research:** Computation of Shapley Values is relevant for more and more application areas most prominently Explainable AI. Black box (game-agnostic) estimation methods are important with better methods proposed every year. Therein, this work contributes to an important research field. The most important aspect of the contribution is the dynamic nature of the estimation, and that it natively supports estimation of current estimation progress (which is often a positive argument brought in favor of [4]).
---
- **Well Written and Good Presentation:** The paper is well written and clear to follow. The contribution is clearly presented and well organized. The presentation good.
---
- **Noation:** The mathematical notation and presentation is good.
---
- **Good Baselines:** The paper compares itself already to the most important baseline methods. While the current SOTA [2] is missing, the current selection is already quite well chosen. Some improvements can be done.
---
- **Code:** I greatly appreciate the submission of the source code and the quality of the code. Thank you for including the demo notebooks they are interesting to go through.

**Weaknesses:**

- **Empirical Evaluation:** While the empirical evaluation is generally well done, it would be nice to also include some of the methods described in the below point in the evaluation such as [1] and [4]. The comparison with [4] is interesting since it is usually also very efficient but a bit more limited in terms of performance and thus serves as a good baseline how hard the problems are and how impressive the gains of improvements are. I also do not like that the empirical evaluation does not compare against *real ground truth* values at least in a couple of cases, but estimates them with KernelSHAP running longer. Real ground truths can be easily achieved by computing via brute force for datasets with a moderate number of features (16 features needs approximately $2^{16} \approx 64k$ model calls). This should be doable for some settings.
---
- **Missing Related Work:** While the paper compares itself to modern methods to estimate the Shapley value empirically, the related work is not very well prepared in the current version of the manuscript and the majority of the related work is moved into the appendix. The whole SIM-SHAP method seems to me very related to [1] and potential follow-up works on this. I feel like this should be described in the manuscript. Some other impactful Shapley value estimation methods ([3] and [4]) are also missing. Most prominently, the RegressionMSR method proposed in [2] would need to be discussed in the work, since it presents the current state of the art in terms of Shapley value estimation. Since the paper was only released rather recently, I don't think an empirical evaluation is necessary (of course interesting) but delineating against is would definitely improve the work.
---
- **Doubts about LeverageSHAP usage:** I have my doubts about LeverageSHAP not being able to be used on classification tasks and the notion of it being imputer-dependent since LeverageSHAP is a black-box estimation method. While the *implementation* of LeverageSHAP may be limited to these tasks, it should be easily transferrable. Also see question 1.
---
For me this work is **quite borderline** (in its current state sitting a bit on the reject side), but depending on the resolution (or partial resolution) of my concerns, I think **this paper could be accepted.**

**Questions:**

### Question 1
You write
> SIM-Shapley uniquely provides both game and imputer agnosticism—capabilities absent in existing methods

and
> Leverage- SHAP (Musco & Witter, 2025) employs statistical leverage scores to improve sampling efficiency but is limited to mean-value imputation and specific task types

How is LeverageSHAP **not** imputer invariant? LeverageSHAP is basically KernelSHAP but a smarter coalition selection / sampling. The imputer used behind the value function does not influence the applicability of LeverageSHAP, no?

---
### Question 2
How does your method compare to [1]?

---

> ### Author Response · Authors · 2025-11-19
>
> **Dear Reviewer ZozL,**
>
> Thank you very much for your insightful review.Below, we provide a point-by-point response to further clarify the work and address all concerns. The proposed revisions and additions have been incorporated into the revised manuscript (marked in blue).
>
> ### Weaknesses
>
> 1. **Empirical Evaluation**: The related works you mentioned—[1], [3], and [4]—are indeed relevant but do not belong to the class of **Regression-based methods**, which correspond to Eq. (6) in our paper. Following previous works proposing the regression formulation and noticing that they didn't compare these early existing baselines, we consider it is not necessary and less relevant to compare the proposed regression-based SV method with all improved sampling techniques. Specifically,the original KernelSHAP paper compares only with simple permutation sampling rather than advanced variants; In the presentation of LeverageSHAP, they only compare with KernelSHAP and improved techniques of KernelSHAP. In this work, we benchmark against widely used and recent baselines, which sufficiently demonstrates effectiveness.
>
> 2. **Missing Related work:** Regarding RegressionMSR [2]: we fully agree that it is a strong regression-based method. However, **its publication date was too close** to ours to appear during our literature review, and what more imporantly, it has **no open-source implementation is currently available**. Therefore we regret that at current stage it is not possible for us to include additional experiments for this method. However, we have now mentioned it in the background section *Related Work* in **Appendix A** to raise interesting audience's attention and we hope in the future we could further followup once it has public code implmentation.
>
>
> 3. **Doubts about LeverageSHAP**: Many prior works (see below) focus on a single value function, however, value functions are *not* the core contribution of our paper. Our goal is to ensure **fair comparisons under the same value function**, which reflects algorithmic efficiency. Many methods do not allow easy modification. In the following, we list some techinical challenges regarding why modifications is not an easy effort:
>
>     a) FastSHAP embeds the value function into its NN training, changing it often breaks the code
>     b) LeverageSHAP hard-codes regression-only outputs with a continuous value function. However, for classification problems, Shapley values need label-specific results—e.g., SHAP’s ImageNet example (https://github.com/shap/shap).
>     c) The paper of LeverageSHAP mentions nothing about imputer and it fixes a mean-value filling strategy in the logic of code.
>     Conversely, KernelSHAP and SIM-Shapley treat value functions and imputer as independent Python classes, making both regression and classification tasks naturally supported with arbitrary imputers.
>
> To address the need for real ground truth values, we employ unbiased KernelSHAP, running it until a stringent threshold (defined in equation 11) is met. This threshold, representing the iterative value change, ensures Shapley value stabilization and a precise approximation of the ground truth when a quite low threshold obtains. This approach, previously validated in ICLR-accepted FastSHAP and LeverageSHAP, obviates the necessity for obtaining the exact ground truth, particularly given our hardware limitations. In **Appendix E.3**, we add two supplementary experiments on this threshold corresponding to bias and we hope this revised version will be helpful for you.
>
> Sincerely, we hope our comments could well solve your concerns.

---

### Official Review · Reviewer_Jdo3 · 2025-10-29

**Soundness:** 2
**Presentation:** 2
**Contribution:** 1
**Rating:** 2
**Confidence:** 4

**Summary:**

This paper considers the problem of estimating Shapley values of a general value function $v$. They propose a linear regression-based method similar to KernelSHAP. Instead of solving the weighted and constrained least squares problem once, they iteratively solve a sequence of regression problems. When solving the next regression problem, they draw $m$ new samples and add a regularization term on the "moving average" solution of the regression problems so far.

**Strengths:**

N/A

**Weaknesses:**

Big:

1. I think the biggest structural issue in this paper is how they discuss and compare to prior work. Across the discussions and experiments, they consider different subsets of the estimators FastSHAP, SimSHAP, LeverageSHAP, and KernelSHAP. The first issue is that they selectively compare to these algorithms e.g., they'll sometimes ignore LeverageSHAP or FastSHAP depending on the experiment. The bigger issue is that they don't consider other SOTA estimators which include PermutationSHAP, and (two recent methods) RegressionMSR and ProxySPEX.

2. The paper ignores fundamental properties of estimators, or says things that are just incorrect about them:
    * They claim prior estimators can't be used for general value functions $v$, but, except for methods like TreeSHAP (which isn't mentioned in the paper), this is just false. Basically all near-SOTA Shapley value estimators are agnostic to the value function. (I think they know this e.g., they present a "variant" of their estimator for global explanations in Appendix C but the only change is one line about how the value function $v$ is computed.) It is true that *implementations* of these algorithms are generally tied to specific value functions but, I would say a) the onus for modifying the standard implementation is on researchers if they want experiments on a new kind of value function, and b) the shap-iq library is extracting the value function logic to an arbitrary class that makes it easy to switch which value function is used.
    * They treat KernelSHAP and LeverageSHAP as different algorithms. In reality, they're the same algorithm except a) the sampling distribution is different, and b) LeverageSHAP solves a constrained regression problem whereas KernelSHAP uses very large weights on the empty set and full sert. The fact that the experiments in the current paper have such a big gap in performance between these two methods indicates something is strange about the implementation.
    * FastSHAP is fundamentally a different *kind* of estimator because it learns a neural network to predict the Shapley values with respect to *multiple* value functions. So if you're only estimating the Shapley value of a single value function, it doesn't really make sense to use FastSHAP. (To be fair, this is mentioned in passing in a runtime table in the experiments.)

3. As for their algorithm, I'm confused bordering on concerned. Instead of solving one regression problem, they iteratively solve a sequence of algorithms. Intuitively, I don't see the value in distributing the total budget into a sequence of worse solutions than just solving the regression problem once (but hey, I could be wrong about this). The bigger concern I have is that, as written, Algorithm 1 is given a budget of $m$ (like the other estimators), but then makes $m$ independent samples for each of the $T$ iterations. Effectively, this algorithm is getting to see $T * m$ evaluations of the value functions, whereas the algorithms they compare to gets only $m$ evaluations. If this is how the experiments are implemented, they're deeply unfair because SimShapley is getting way more access to $v$ than the other algorithms.

4. As mentioned earlier, their experiments are inconsistent in terms of which algorithms they compare to. And they present performance on different datasets in different ways. Pessimistically, this would indicate cherrypicking of experimental results. My **strong** suggestion is to standardize their experiments: Plot bias as a function of sample size in one big plot with one subplot per dataset. If you want, make a similar plot for time, and a similar one for "global" value functions (again, the only difference here is how $v$ is defined so, even if it means modifying some implementation code, you should compare to *every* Shapley value estimator that you mention). This would be a) more visually appealing, and b) much easier to see overall performance of each estimator in a fair way.

Here are the recent (very good) Shapley value estimators you should compare to:

ProxySPEX: https://arxiv.org/abs/2505.17495

RegressionMSR: https://arxiv.org/abs/2506.11849

Minor:

* There's already an algorithm called "SimSHAP" so calling this one "SimShapley" seems confusingly similar. Especially because this algorithm seems to be quite different from "SimSHAP".

**Questions:**

* Is there a typo in Equation 7b? I.e., should it be $\delta^{(n+1)} = t \beta^{(n)} + (1-t) \delta^{(n)}$.

* In Table 2, you run KernelSHAP with $m=64$ and SimShapley with $m=64$ and some $T$ (I assume greater than 1). How many evaluations does each algorithm get? Is it that KernelSHAP gets $64$ and SimShapley gets $64T$? What do you set $T$ to?

---

> ### Author Response · Authors · 2025-11-19
>
> **Dear Reviewer Jdo3,**
>
> Thank you for your thoughtful and constructive comments. We address your concerns below. The proposed revisions and additions have been incorporated into the revised manuscript (marked in blue).
>
> 1. Regarding the additional baseline estimators mentioned: ProxySPEX and RegressionMSR are indeed interesting works. However, their publication dates are extremely close to (or later than) our submission, therefore they were unavailable during our literature review. In addition, ProxySPEX leverages sparse fourier-based approximator and focuses primarily on **LLM explanations—a very different setting from ours**—and RegressionMSR **does not yet provide an open-source implementation**. Our comparisons therefore focus on traditional and widely adopted regression-based methods. To record this gap, we have mentioned in the section of related work in **Appendix A**.
>
> 2. Concerning value functions: many prior works focus on a single value function, but value functions are *not* the core contribution of our paper. Our goal is to ensure **fair comparisons under the same value function**, which reflects algorithmic efficiency. In practice, modifying value functions in several baselines is highly nontrivial:
>
> - In FastSHAP, the value function is embedded into the neural-network training loss; changing it often breaks the code.
> - LeverageSHAP hard-codes its value function and only supports regression; classification Shapley values require label-specific outputs (e.g., as in SHAP’s VGG16 ImageNet example, https://github.com/shap/shap).
>
>     By contrast, KernelSHAP (also in shap-iq) and SIM-Shapley treat value functions as separate classes, making modification feasible. After examining shap-iq’s code, we found that many of our baselines (except KernelSHAP) are not implemented, so we rely on available source code.
>
> 3. On performance differences: LeverageSHAP and SIM-Shapley both build upon KernelSHAP with additional improvements, so a performance gap is expected. For fairness, we evaluate KernelSHAP and LeverageSHAP using the official repository of LeverageSHAP [1], which reproduces consistent results with the original paper. FastSHAP, while one-shot, is still regression-based; comparing its strengths and limitations with sampling-style improvements is meaningful.
>
> 4. Given the significant heterogeneity of baselines, fully unifying comparison standards is nearly impossible. Thus, we provide multiple perspectives to help readers understand the trade-offs. KernelSHAP—being the foundation of many baselines—is always included for reference.
>
> Finally, we desire to answer your questions. Regarding sample size consistency: in KernelSHAP (and shap-iq’s implementation), LeverageSHAP, and SIM-Shapley, we utilize batch sizes per iteration; e.g., if the total sample size is 2048 with batch size 512, then the number of iterations is 4 (T × B = M), ensuring consistent sample sizes across methods. As for the equation 7a, \beta^{n+1} is correct since \beta is the final estimator and \delta is obtained just depending on the sampling in each iteration.
>
> In summary, we Sincerely hope our comments could well solve your concerns and we would greatly appreciate it if you could recognize our work。
>
> **References**
> [1] Christopher Musco and R. Teal Witter. *Provably Accurate Shapley Value Estimation via Leverage Score Sampling*, 2025.

---

### Official Review · Reviewer_LoUX · 2025-10-30

**Soundness:** 3
**Presentation:** 3
**Contribution:** 2
**Rating:** 2
**Confidence:** 5

**Summary:**

This paper introduces SIM-Shapley (Stochastic Iterative Momentum for Shapley values), a new method for approximating Shapley values, which are a key tool for feature attribution in explainable AI (XAI). The primary challenge with Shapley values is their exponential computational complexity, making them impractical for many real-world applications. The authors propose SIM-Shapley to address this by reformulating the Shapley value calculation as a stochastic optimization problem.

**Strengths:**

1.	The proposed method covers both local and global explanations.
2.	Clear reformulation of KernelSHAP as a constrained stochastic optimization with a simple EMA update and a closed-form per-iteration solution.
3.	The method stays model-agnostic.

**Weaknesses:**

1.	The authors claim that their method fundamentally re-conceptualizes SV computation. However, the core of their process is a series of mature stochastic optimization techniques.
2.	The early stopping method (Eq.11) is heuristic and has no sensitivity analysis to the parameter epsilon.
3.	The choice of the parameter xi in Eq. 12 is crude and lacks analysis.
4.	The authors claim that the method can be used for Shapley Interactions, but provide no experiments to demonstrate this.

**Questions:**

1.	Line 159: The constraint in equation 5 should preserve the efficiency property of SV.
2.	The main convergence theorem targets the SIM-Shapley fixed point $\beta^*$ that depends on $\lambda$, not the unregularized KernelSHAP target $\beta$. Hence, the theory guarantees a fast approach to $\beta^*$rather than a fast approach to $\beta$. Figure 1 suggests a small bias for small λ, but there is no explicit bound for $‖beta^*−\beta‖$ or a rate as $\lambda \to 0$.
3.	The proof of Theorem 1(Appendix B.1) seems to rely on a key assumption $mathrm{Var}(\delta^{(j)})\le \mathrm{Var}(\beta)$, which is not always right.

---

> ### Author Response · Authors · 2025-11-19
>
> **Dear Reviewer LoUX,**
>
> Thank you for your feedbacks. Below, we provide a point-by-point response to further clarify the work and address all concerns. The proposed revisions and additions have been incorporated into the revised manuscript (marked in blue).
>
> ### Weaknesses
>
> 1. We agree that the phrase *“fundamentally re-conceptualize”* is indeed a bit too strong. We have refine the wording to a more accurate description in our revised version (third paragraph of Background section). Nevertheless, following the ICLR reviewer guidelines—*“Submissions bring value when they convincingly demonstrate new, relevant, impactful insights (empirical, theoretical, or practical)”*—we believe our work meaningfully integrates modern numerical techniques with the classical sampling pipeline, which may provide new practical insights for the community.
>
> 2. &3. Regarding the hyperparameters in Eq. (11) and Eq. (12): these serve as empirical engineering choices that control the convergence criterion (i.e., the magnitude of value change between iterations). For many non-theoretical papers, such sensitivity analysis is typically not the central focus. For example, the KernelSHAP paper [1] also adopts iterative convergence without reporting sensitivity results. To solve your concern, we have already add two extra experiments in **Appendix E.3** of our revised pdf and we hope this would be helpful for you.
>
>
> 3. On Shapley Interactions (SIs): SIs group features into players, which is compatible with our framework. However, since our paper focuses on improving the computation procedure for standard SV, omitting SIs experiments is reasonable. Moreover, most baseline papers we compare with likewise do not evaluate higher-order interactions.
>
> ### Questions
>
> 1. Thank you for catching the error in line 159. The SV property established in [2] confirms your observation, and we have corrected this mistake in revised version.
>
> 2. Regarding the theoretical component: analyzing regression-based SV estimators is notoriously difficult—early KernelSHAP works also did not provide full theoretical treatment [3]. Establishing guarantees usually requires additional and sometimes restrictive assumptions. For example as mentioned by Covert et al. [1], achieving stronger guarantees requires introducing extra algorithmic tools, and the resulting properties may deviate from empirical observations (i.e unbiased KernelSHAP converges slower than KernelSHAP).
>
> 3. In our proof, one only needs to focus on the $0$-th iteration step; Eq. (7) reduces to a ridge regression form that reduces variance, which justifies the inequality that you confuse.
>
> We hope the above replies could well solve your concerns and we would greatly appreciate it if you could recognize our work.
>
> **References**
>
> [1] Ian Covert and Su-In Lee. *Improving KernelSHAP: Practical Shapley Value Estimation via Linear Regression*, 2021.
> [2] Anupam Datta et al. *Algorithmic Transparency via Quantitative Input Influence*, 2016.
> [3] Scott Lundberg and Su-In Lee. A unified approach to interpreting model predictions, 2017

---

### Official Review · Reviewer_9Y6m · 2025-11-01

**Soundness:** 3
**Presentation:** 3
**Contribution:** 3
**Rating:** 8
**Confidence:** 5

**Summary:**

This paper proposes SIM-Shapley, a Shapley value approximation method based on stochastic iterative momentum. Its core idea is to reformulate Shapley value computation as a stochastic optimization problem. The key innovations include: (1) achieving linear Q-convergence through Exponential Moving Average (EMA)-based updates and adaptive mini-batch sampling, with a variance contraction rate of $(1-t)^2$; (2) introducing three stability mechanisms: $\ell_2$ regularization, negative-sampling detection, and initialization bias correction; (3) supporting both local and global explanation modes while being game-agnostic and imputer-agnostic. Experiments show that SIM-Shapley is 60%-85% faster than baselines such as KernelSHAP and SAGE across classification, regression, image, and clinical tasks, with a 10%-15% reduction in bias.

**Strengths:**

1. The paper is well-written, with a clear structure that allows readers to easily follow the authors' reasoning.
2. The comparison between SIM-Shapley and other Shapley value computation methods is thorough and clear, enabling readers to readily grasp the paper's contributions.
3. Relevant experiments effectively demonstrate the superiority of the proposed method, particularly in accelerating Shapley value computation.
4. The paper addresses interpretability, a critical issue in deep learning models. By focusing on accelerating Shapley value computation, this work holds significant importance—it facilitates the adoption of deep learning models in high-reliability industries.

**Weaknesses:**

The paper is relatively comprehensive, and there are no major fundamental issues. However, two minor points require attention:
1. **Paper Formatting:** The authors are requested to review the paper's formatting. For instance, on Page 21 of the supplementary materials, some figures/icons clearly exceed the paper's margins.
2. **Evaluation Metrics:** The paper primarily uses the error between estimated Shapley values and the ground truth to evaluate performance. Readers are curious about how the proposed method performs under a broader set of evaluation metrics, such as Deletion/Insertion and mu-fidelity.

**Questions:**

Refer to the "Weaknesses" section above.

---

> ### Author Response · Authors · 2025-11-19
>
> **Dear Reviewer 9Y6m,**
>
> Thank you very much for your valuable feedback.
>
> We are delighted to receive your generous praise for our work. Your positive evaluation is greatly appreciated and serves as a strong encouragement to our team.
>
> We have corrected the formatting issues you pointed out in the revised manuscript, along with several additional improvements and minor fixes (all marked in blue).
>
> Regarding the evaluation metrics, we are grateful for your insightful comments. Indeed, deletion/insertion and μ-fidelity are widely used and meaningful metrics in XAI applications. However, from our perspective, these metrics are typically designed to compare heterogeneous explanation methods. Since our work focuses specifically on improving the computational efficiency and accuracy of Shapley value estimation—and all baselines in our experiments produce Shapley values under the same definition—additional XAI metrics are not strictly necessary for evaluating the quality of SV estimates themselves.
> In future work, as we extend our methods to broader tasks and compare against a wider set of non-SV explanation approaches, incorporating these metrics will be essential, and we appreciate your recommendation.
>
> Finally, we hope that the revised version and our clarifications adequately address your concerns. We would be grateful if you could consider our work favorably.

---

### Meta-Review · Area_Chair_zfjc · 2026-01-05

**Summary:**

The paper develops a technique for computing Shapley values which is a popular tool for producing feature attributions. The approach takes standard tools from stochastic optimization and applies them to the shapley value computation recast as a stochastic optimizaiton. While the approach is different, there was a consensus among the reviewers that the approach was overstated and the qualities of the existing approaches were understanded (e.g, the reviewers note that existing methods can work with different value functions or tasks even if the released code is for a single one like classification). In all, it looks like the work needs to sharpen these points to be ready for publication

**Reviewer Concerns:**

9Y6m - liked the paper. no real concerns

LoUX - concerns about the novelty of using stochastic optimization and crude analysis

Jdo3 - concerns about the experiments and the prior estimators and general value functions

ZozL - the empirical evaluation and related work

**Reviewer Scores:**

I don't think any of the reviews would change mostly because of improvement needed to the baseline. Like "breaks the code" would not be strong enough to rule out a baseline from doing thing. Most AI code released for papers is for tasks in the paper not tasks the math could do

---

### Decision · Program_Chairs · 2026-01-26

Reject